# Group-level cooperation in chimpanzees is shaped by strong social ties

Liran Samuni [1,2,3✉], Catherine Crockford[1,2] & Roman M. Wittig [1,2]

Humans maintain extensive social ties of varying preferences, providing a range of opportunities for beneficial cooperative exchange that may promote collective action and our unique capacity for large-scale cooperation. Similarly, non-human animals maintain differentiated social relationships that promote dyadic cooperative exchange, but their link to cooperative collective action is little known. Here, we investigate the influence of social relationship properties on male and female chimpanzee participations in a costly form of group action, intergroup encounters. We find that intergroup encounter participation increases with a greater number of other participants as well as when participants are maternal kin or social bond partners, and that these effects are independent from one another and from the likelihood to associate with certain partners. Together, strong social relationships between kin and non-kin facilitate group-level cooperation in one of our closest living relatives, suggesting that social bonds may be integral to the evolution of cooperation in our own species.

[1] Max Planck Institute for Evolutionary Anthropology, 04103 Leipzig, Germany. [2] Taï Chimpanzee Project, CSRS,  01 BP 1303 Abidjan, Ivory Coast. [3] Department of Human Evolutionary Biology, Harvard University, 02138 Cambridge, MA, USA. ✉email: liran_samuni@eva.mpg.de

Enduring and strong non-reproductive social ties that are formed between unrelated adults (i.e., social bonds) are critical components to the fitness, health and well-being of not only humans[1,2] but many non-human mammals as well[3–9]. A defining feature of social bonds is cooperative exchange[10], likely facilitated by repeated interactions between partners which may stabilize direct and indirect reciprocity[11–13]. Human social relationships, from bond partners to more distant associates, form a continuum of extensive social networks characterized by social ties of varying preferences and degrees of reciprocal exchange[14,15]. These properties of human social networks are theorized to drive not only dyadic cooperation, but as well to facilitate collective action and unique large-scale cooperation[14], hallmark of human societies. Other group living animals also maintain differentiated social relationships with kin and non-kin[3,5,6,8–10,16–21], and engage in collective action and group-level cooperative acts with in-group members[22–27]. Whereas differentiated social relationships in non-human animals may promote dyadic cooperative tasks associated with fitness benefits, like defence efficiency against predators or threats[19,28–30] or alloparental care[20,31,32], little empirical evidence exists that can address if and how these relationships influence group-level cooperation and collective action.

Intergroup competitive interactions involve group-level acts that rely on cooperation and collective action in humans[33,34] but as well in one of humans' closest relatives, chimpanzees[22,23]. Although intergroup competition may incur fitness benefits in both species[33,35,36], it has high uncertainty in returns. Furthermore, intergroup competition may result in substantial costs for the individual and group, for example via the loss of a territory or reduced survival of group members[35,37,38]. The cost-to-benefit ratio associated with the conflict can be optimized by a cohesive and cooperative response of group members[22,23,39]. Despite the prominent role of the maintenance of a range of differentiated relationships in cooperation in humans, little is known about how these may operate during risky intergroup competitive interactions. We therefore investigate here how differentiated social relationships and social preference may influence the decision to contribute to intergroup competitive interactions in one of our closest living relatives, chimpanzees.

Following the 'imbalance of power' hypothesis[39] the odds to prevail in intergroup competitive interactions (hereafter intergroup encounters) are likely heavily reliant on strength in numbers. Thus, and in accordance with previous studies (chimpanzee[40]; lion[41]), we predict that when more other in-group members participate in the intergroup encounter the chance for a specific individual to join increases. Still, prior assessment of number superiority is limited, making success odds of intergroup encounters highly uncertain[23]. Thus, it appears that participation of many individuals (i.e., cohesive group engagement) alone may not be sufficient in facilitating contribution to intergroup encounters. Crucially, we hypothesize in addition that a mechanism that reduces the likelihood of defection and increases predictability of reciprocity in dangerous situations may drive participation. Strong social relationships in humans, chimpanzees, and other social animals, such as kin relations and social bonds, are associated with increased reciprocal exchange that is stable over time[10]. In chimpanzees, maternal kin and social bond partners more frequently support each other during within-group agonistic interactions, groom, and share food[16,17,42,43]. It is reasonable to assume that such consistent social exchange that occurs within one's group and aids in navigating within-group competitive interactions will also predict coordination and coalitionary support during between-group competitive interactions[44], potentially facilitated by oxytocinergic system activity[45,46]. This hypothesis is supported by studies showing that chimpanzee males more often patrol the borders of their territory together with their maternal brothers compared to unrelated dyads[18] or with males with which they groom more[44], and that the presence of a kin or non-kin bond partner during chimpanzee intergroup encounters appears to buffer stress responses[4]. Therefore, we predict that the presence of social bond partners and maternal kin would drive participation. In concert, examining the hypotheses that the number of participants and social preference influence the decision to take part in hostile encounters with neighbours may potentially reveal whether numerical strength and/or partner choice are mechanisms through which chimpanzees optimize the cost-to-benefit ratio associated with such conflicts to promote cooperation.

To examine the effects of social properties such as the number of participating group members, or social preference towards kin and social bond partners on the emergence of cooperation we determine how these affect chimpanzee participation decisions in intergroup encounters. We investigate this during intergroup encounters that were either initiated or involved an active approach behaviour by the in-group, as otherwise no voluntary participation was made. Thus, we can use the identity of individuals present during the encounter as a proxy of a pre-encounter decision-making process (see the "Methods" section). To do so, we use 55 cumulative observation years of demographic, genetic and behavioural data on 36 males and 75 females from three groups within the Taï Forest, Côte d'Ivoire. Although there are clear male-biased participation predisposition across chimpanzee populations[23,40,44,47], in Taï, females are active participants in the majority of intergroup interactions[23,45,48] and are involved in aggressive displays and coalitionary attacks against out-group individuals[48]. Following our predictions for the two hypotheses, we test the effect of (i) the number of other male and female in-group participants ('strength in number hypothesis'), and (ii) the presence of adult maternal kin or social bond partners ('predictability of coalitionary support hypothesis') on the likelihood to contribute to intergroup encounters in both sexes. We define maternal kin as mother–offspring and grandmother–grandoffspring dyads or maternal siblings, and social bond partners as dyads with stable, mutual, and preferred grooming relationships (see the "Methods" section).

Across animal species, intergroup participation tendencies are likely influenced by a concert of social and ecological variables, as well as individual attributes[22,24–26,40,41,49,50]. Thus, to reliably evaluate the effect of our hypothesis testing predictors, we control in our analyses for additional set of predictors. We measure association patterns to control for spatial proximity amongst participants driving participation likelihood, and also to account for another characteristic of social relationships, geographic decay (i.e., the chance that with increased distance between individuals there would be a decrease in social ties and hence decreased participation and vice versa). In addition, we consider some of the potential costs and benefits which previously have been shown to explain participation tendencies across non-human primate species. For instance, intergroup encounter participation is associated with energetic costs and high risk of injury. Thus, as indicators of physical condition we account for individuals' age and dominance rank[22,24,26], food availability[25,26], and females' number of days into gestation and early lactation periods[25,26]. Intergroup encounter participation may also result in reproductive and nutritional benefits to all group members, regardless of participation. Thus, we account for potential inclusive fitness gains by including the number of all living maternal kin in the group (hereafter community), the number of living offspring of male participants[22,25,49] and the presence of a son approaching reproductive age for female participants (see the "Methods" section).

In this work, we find that cohesive engagement of many community members and the presence of adult maternal kin or

social bond partners predict the likelihood of males and females to contribute to hostile encounters with rival groups. These results are independent of potential costs and benefits of intergroup encounter participation, like age, dominance rank, the number of offspring in the community, or the reproductive status of females. Overall, strength in numbers and predictability of reciprocity are associated with cooperation emergence in chimpanzee costly collective acts, providing support for strong and enduring social bonds promoting group-level cooperation not only in humans, but also chimpanzees.

## Results

**Intergroup encounters in Taï chimpanzees.** We observed a total of 186 intergroup encounters in North (average of 8.85 per year), 166 in South (average of 8.73 per year) and 139 in East (average of 13.9 per year). Most intergroup encounters were vocal, and contact encounters comprised 35% of all encounters. We observed lethal aggression on ~1% of intergroup encounters, none of which targeted fully grown individuals. Overall, we obtained complete information on the identity of all participants, their reaction, and the type of encounter for 403 out of the 491 encounters (82%), and 343 out of the 403 cases involved an active reaction of in-group individuals towards the out-group (e.g., patrol towards, vocalize, attack). For the remaining 343 encounters, at least one male participated in 96% of cases, as opposed to 90% in females. When accounting for all adult individuals in the community as potential participants, 86% and 48% of all males and females participated, respectively. The average (±SD) number of participants per intergroup encounter was 3.43 ± 1.54 in males (ranging 0–7 males) and 5.03 ± 3.57 in females (ranging 0–18 females).

**Factors predicting intergroup encounter participation.** We investigated variation in participation likelihood of male and female chimpanzees. The 'male participation' and 'female participation' full-null model comparisons were significant (generalized linear mixed model (GLMM) likelihood ratio test: male participation— $\chi^2 = 37.578$, df = 4, $p = 1.37e-7$, $R^2_{marginal} = 0.22$, $R^2_{conditional} = 0.45$; female participation— $\chi^2 = 234.874$, df = 4, $p = 1.17e-49$, $R^2_{marginal} = 0.47$, $R^2_{conditional} = 0.61$; Tables 1 and 2).

The number of other male and female participants significantly and positively affected males and females' participation likelihood ('male participation' GLMM: $p = 0.001$ other females and $p = 0.023$ other males; 'female participation' GLMM: $p = 4.27e-30$ other females vs. $p = 0.0017$ other males). Participation likelihood of both sexes gradually increased with more female participants, but the effect of the number of male participants differed between the sexes (Fig. 1). While participation likelihood of females gradually increased with more male participants, for males it was the presence of at least one other male, rather than a gradual increase in the number of males, that strongly influenced their participation likelihood. For both sexes, we observed the steepest increase in participation probabilities with more same-sex participation.

Whereas the number of living maternal kin in the community had no clear significant effect on male participation (GLMM: $p = 0.609$), the presence of adult maternal kin in the encounter positively predicted their participation ($p = 0.005$; Fig. 2a). In females, the total number of living maternal kin in the community had a significant negative effect ($p = 0.0026$) on female participation likelihood, however, both the presence of an adult maternal kin in the encounter ($p = 1.02e-8$; Fig. 2a) and a son approaching reproductive age (>8 years) in the community ($p = 0.0009$) had a significant positive effect on the likelihood of a female to participate. Social preference had a significant effect on participation likelihood of both sexes (females: $p = 1.98e-12$; males: $p = 0.0026$; Fig. 2b), such that both males and females had

higher participation tendencies when their social bond partner participated as well. Spatial proximity, measured by the average dyadic association scores with all other participants, positively influenced participation probabilities of females ($p = 2.27e-14$; Fig. 3a) and males ($p = 0.0017$; Fig. 3b).

The number of gestation days of females on the day of an encounter had a significant negative impact (GLMM: $p = 0.0176$) on their participation which decreased with proximity to parturition, and females that showed maximal sexual swelling were significantly more likely to participate in encounters ($p = 2.85e-9$) in comparison to females that did not show maximal swelling. Furthermore, females were more likely to participate in contact versus vocal intergroup encounters ($p = 0.016$).

The age and dominance rank of females and males, the type of intergroup encounter for males and their paternity success (number of living offspring), presence of a young infant (<2 years) in females, and general monthly fruit availability, had no clear significant effects on participation likelihood.

## Discussion

Within-group cohesive engagement of males and females, relatedness, and social preference each have a strong, independent influence on the participation of male and female chimpanzee in one of the costliest forms of group actions, hostile intergroup interactions. We found that participation disposition of males and females increased when more other in-group members participated, both same and opposite sex, supporting the hypothesis that 'strength in numbers' facilitates participation. Although cohesive engagement of many community members was influential, it was not the only participation driver. Additionally, we found that the decision to contribute to the group act heavily depended on the identity of other participants as maternal kin or social bond partners, types of relationships associated with more predictable reciprocity[16,17,42,43].

Both cohesive engagement of many community members and of specific preferred partners likely encompasses the range of social relationships within one's group, from weakly to strongly bonded, facilitating coordinated contribution to intergroup competitive interactions. These effects were independent from potential energetic and reproductive costs of intergroup encounter participation observed in chimpanzees and other primates, such as old age, low dominance rank, gestation timing, and lactation[22,25,26,49]. Furthermore, the positive effect of relatedness and social preference on participation tendencies occurred irrespective of another social aspect which may drive interaction likelihood[14], spatial proximity (association degree) between participants. This, in addition to using only active intergroup encounters, further indicates the observed pattern is not a byproduct of association tendencies and is independent of the likelihood to be present with others who are at the encounter area. Our results echo findings of studies on hunter-gatherer societies that suggest strong social ties, whether kin or non-kin relations, facilitate the emergence of cooperation[14].

The degree of genetic relatedness to other community members is expected to influence individuals' contribution to acts that are potentially costly, as participants are bound to gain higher indirect fitness benefits with more relatives in the community[51]. Higher numbers of maternal kin in the community likely increases the probability of acting with adult kin during the intergroup encounter making it difficult to distinguish whether it is overall relatedness or direct interactions with kin that is linked with cooperation emergence. Testing the effect of the presence of adult maternal kin in the encounter while accounting for the number of maternal kin in the community, allows us to examine the mechanisms through which genetic relatedness may operate

**Table 1 'Female participation' GLMM testing the effect of strength in numbers and predictability of coalitionary support on participation likelihood (N = 343 encounters, 75 adult female subjects).**

| Term | Reference category | Estimate | SE | 95% CI | Chisq | p |
|---|---|---|---|---|---|---|
| Intercept | | −3.403 | 0.165 | −3.723, −3.100 | – | – |
| *Test predictors* | | | | | | |
| Maternal kin (present) | Absent | 1.529 | 0.284 | **0.976, 2.206** | 32.795 | **1.02e−8** |
| Bond partner (present) | Absent | 0.890 | 0.129 | **0.651, 1.159** | 49.502 | **1.98e−12** |
| Number females[a] | | 1.402 | 0.082 | **1.255, 1.568** | 129.918 | **4.27e−30** |
| Number males[b] | | 0.229 | 0.071 | **0.089, 0.367** | 9.843 | **0.0017** |
| *Control predictors* | | | | | | |
| Son >8 years (yes) | No | 0.680 | 0.208 | **0.272, 1.085** | 10.931 | **0.0009** |
| Number kin[c] | | −0.356 | 0.113 | **−0.592, −0.132** | 9.065 | **0.0026** |
| Dyadic association[d] | | 0.794 | 0.083 | **0.633, 0.968** | 58.284 | **2.27e−14** |
| Dominance rank[e] | | 0.163 | 0.089 | −0.012, 0.335 | 2.564 | 0.109 |
| Age[f] | | −0.276 | 0.149 | −0.556, 0.014 | 3.362 | 0.067 |
| Age[f]—squared | | −0.141 | 0.127 | −0.383, 0.088 | 1.228 | 0.268 |
| Reproductive status (fully tumescent) | Non-cycling | 1.221 | 0.213 | **0.826, 1.636** | 35.278 | **2.85e−9** |
| Gestation days[g] | | −0.131 | 0.055 | **−0.237, −0.020** | 5.636 | **0.0176** |
| Infant <2 years (yes) | No | 0.134 | 0.120 | −0.361, 0.087 | 1.255 | 0.263 |
| Type (vocal) | Contact | −0.233 | 0.097 | **−0.418, −0.049** | 5.810 | **0.016** |
| Food availability[h] | | 0.041 | 0.061 | −0.084, 0.159 | 0.446 | 0.504 |
| South group | East group | −0.474 | 0.190 | **−0.845, −0.100** | 44.976 | **1.71e−10** |
| North group | East group | 1.541 | 0.264 | **1.063, 1.991** | 44.976 | **1.71e−10** |

In bold appear the 95% CIs that do not overlap 0.
The coded levels are indicated in parenthesis.
[a–h]z-transformed, mean ± SD of the original variables: [a]5.03 ± 3.57, [b]3.43 ± 1.54, [c]1.51 ± 1.02, [d]0.51 ± 0.19, [e]0.55 ± 0.29 (range 0-1 with 1 being the highest social rank), [f]26.67 ± 8.69, [g]11.56 ± 40.25, [h]1.96 ± 1.59.

**Table 2 'Male participation' GLMM testing the effect of strength in numbers and predictability of coalitionary support on participation likelihood (N = 342 encounters, 36 adult male subjects).**

| Term | Reference category | Estimate | SE | 95% CI | Chisq | p |
|---|---|---|---|---|---|---|
| Intercept | | −1.141 | 0.329 | −1.733, −0.364 | – | – |
| *Test predictors* | | | | | | |
| Maternal kin (present) | Absent | 1.177 | 0.455 | **0.387, 2.425** | 7.701 | **0.005** |
| Bond partner (present) | Absent | 0.747 | 0.247 | **0.281, 1.321** | 9.092 | **0.0026** |
| Number females[a] | | 0.553 | 0.132 | **0.355, 0.822** | 11.504 | **0.001** |
| Number males[b] | | 0.449 | 0.146 | **0.179, 0.747** | 5.134 | **0.023** |
| *Control predictors* | | | | | | |
| Dyadic association[c] | | 0.485 | 0.147 | **0.243, 0.771** | 9.809 | **0.0017** |
| Number offspring[d] | | −0.073 | 0.186 | −0.407, 0.313 | 0.027 | 0.870 |
| Dominance rank[e] | | 0.205 | 0.144 | −0.094, 0.470 | 1.575 | 0.210 |
| Age[f] | | −0.272 | 0.335 | −0.846, 0.264 | 0.361 | 0.548 |
| Age[f]—squared | | 0.093 | 0.199 | −0.277, 0.481 | 0.068 | 0.795 |
| Type (vocal) | Contact | −0.124 | 0.193 | −0.536, 0.241 | 0.405 | 0.525 |
| Food availability[g] | | −0.002 | 0.095 | −0.184, 0.185 | 0.001 | 0.986 |
| South group | East group | −0.203 | 0.417 | −0.938, 0.559 | 2.511 | 0.285 |
| North group | East group | 0.838 | 0.530 | −0.130, 1.981 | 2.511 | 0.285 |

In bold appear the 95% CIs that do not overlap 0.
The coded levels are indicated in parenthesis.
[a–f]z-transformed, mean ± SD of the original variables: [a]5.28 ± 3.53, [b]3.04 ± 1.29, [c]0.51 ± 0.18, [d]2.35 ± 2.38, [e]0.69 ± 0.26 (range 0–1 with 1 being the highest social rank), [f]21.31 ± 8.91, [g]1.81 ± 1.37.

in this context, whether directly or indirectly. We found that the total number of living maternal kin of all ages in the community had a negative significant effect on female participation tendencies. Since this number mostly comprised non-adult offspring of females, including offspring that are not carried by their mother, this negative effect may reflect increased energetic and/or injury-related participation costs for weaned young offspring, and hence for their mothers. For males, the number of living maternal kin had no significant effect on male participation tendencies, was in line with findings on male chimpanzees' participation in territorial border patrols in Ngogo[22]. Conversely, the presence of adult maternal kin

in the encounter increased participation of both sexes. For all males, the adult maternal kin present in intergroup encounters were females, predominantly their mothers (one male's maternal kinship was solely his sister), whereas for females, the adult maternal kin present in the intergroup encounter were mainly their sons (two females had both adult daughter and son in the community). Congruent with studies on humans[52], our results suggest that it is the coordination with adult kin, rather than the overall degree of relatedness with community members, that promotes cooperation.

Chimpanzee intergroup encounters are energetically demanding, and contact intergroup encounters are riskier and associated

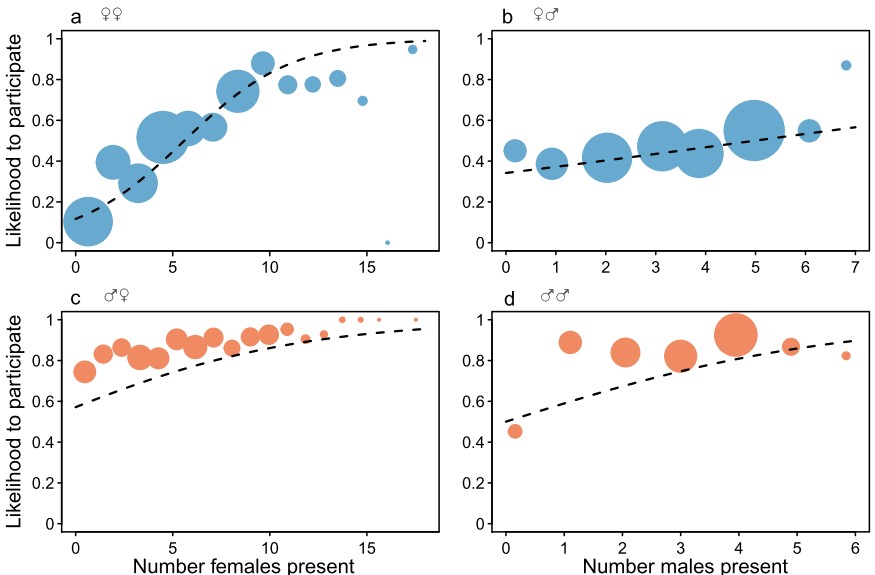

**Fig. 1 Does 'strength in numbers' influence participation?** The effects of the presence of number of other females (**a** and **c**) and males (**b** and **d**) on the likelihood to participate in intergroup encounters in females (blue, 75 individuals) and males (orange, 36 individuals). Larger point denotes larger number of observations and dashed lines represent the model lines. Source data are provided as a Source Data file.

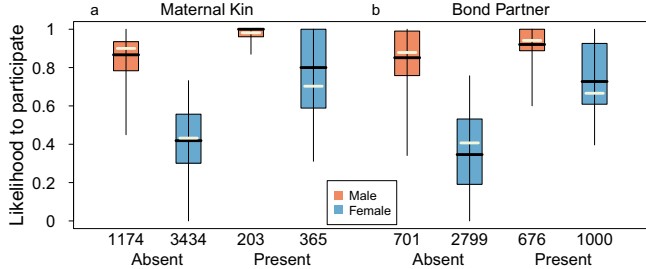

**Fig. 2 Does predictability of reciprocity influence participation?** The effect of the presence of **a** adult maternal kin or **b** social bond partner on the likelihood to participate in intergroup encounters in males (orange, 36 individuals) and females (blue, 75 individuals). The black horizontal lines represent medians, the boxes represent quartiles, the vertical lines represent the 5th and 95th precentiles, and the white horizontal line represent the fitted model. Source data are provided as a Source Data file.

with elevated cortisol levels in comparison with vocal ones[53]. In support of potential energetic costs, time into gestation reduced female likelihood to participate in intergroup encounters. Also, although contact intergroup encounters involve higher risk of injury, we found that they were positively associated with female participation. These results are in line with a previous finding in mountain gorillas, that more aggressive intergroup encounters included a higher proportion of participating in-group members[25]. Across all intergroup encounters, males participated at high rates (86%). Thus, increased female participation translates to more collective engagement of the community as a whole[23]. As cohesive engagement is a key component of both successful intergroup encounter outcome and reduced risk of severe injury[37,41], contact intergroup encounters have a higher likelihood to occur at times of heightened female involvement (indicating heightened overall community involvement) because the odds to prevail are higher and chances to suffer costs are reduced.

Due to the female-dispersal male-philopatry society of chimpanzees, mother–daughter associations at adulthood are rare, and

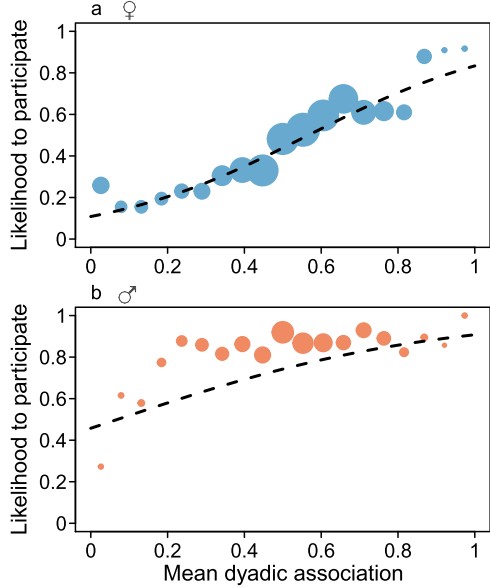

**Fig. 3 Does spatial proximity influence participation?** The effect of average dyadic association with all adult participants on the likelihood of **a** females (blue, 75 individuals) and **b** males (orange, 36 individuals) to participate in intergroup encounters. Larger point denotes larger number of observations and dashed lines represent the model lines. Source data are provided as a Source Data file.

mothers have no direct influence on their daughters' fitness post dispersal. This is not the case for male offspring. Mother–son cohabitation during reproductive ages of sons has the potential to improve the reproductive success of sons, for instance in bonobos[54] and in this population of chimpanzees[55]. Here, we showed that females with sons approaching reproductive age in the community were more likely to participate in intergroup encounters in comparison to females with no such sons in the community. If joint participation increases the chance of gaining improved access to mating opportunities for male community

members[22,35], then mothers who participate in intergroup encounters have the potential to provide future benefits to their sons. We cannot rule out that increased participation of mothers is driven by their sons' increased motivation to participate rather than their own. Nevertheless, regardless of mothers' motivation to participate, the potential benefits of improved future access to mating opportunities to sons remain the same.

Following male biased predisposition for intergroup conflict participation in some primate species, including chimpanzees and humans[23–25,44,56], we observed clear sex-specific differences in participation frequencies. In comparison with males, females showed lower participation rates with larger variation. Further, in accord with homophily predictions for cooperation appearance (i.e., similarity of individuals' attributes, such as sex, increases the likelihood of tie formation[14]), the number of other female or male participants more strongly affected same sex participation. Despite differences in participation rates, similar mechanisms appear to regulate participation of both sexes in territorial defense.

The importance of maintaining strong social networks that encompass a range of relationships for the emergence of cooperation is evident in humans[14,15]. Intergroup competition is known to shape human social group dynamics and cooperative capacities[34,57]. This relationship likely stems from the high stakes associated with participation in intergroup encounters, together with the benefits that one may gain in case of success, where success rests upon the number and value of cooperators. Here we demonstrated the link between intergroup competition, group social dynamics and cooperation in male and female chimpanzees. It may be that the access to and maintenance of differentiated social relationships between kin and non-kin also supports other group-level activities or even energetic demands, as suggested for humans[15,58]. For instance, the maintenance of a range of social ties in humans is thought to support our costly life history and demanding foraging niche[15,58]. Therefore, future research into the effect of differentiated social relationships on chimpanzee capacity for coordinated hunting[59,60] or support of life history trajectories could have an immense potential in revealing the evolutionary processes leading to distinctively human large-scale cooperation that extends beyond borders.

To conclude, cohesive group engagement, interactions with kin, spatial proximity, and social preference all independently contributed to chimpanzee intergroup encounter participation, a risky group act. By choosing to act with maternal kin and social bond partners, and when more in-group members are present, individuals are likely to minimize the costs and maximize the benefits associated with intergroup encounter participation. This emphasizes the importance of chimpanzee social relationship properties and partner choice in driving not only dyadic cooperation, but as well cooperation on the group-level. Our results suggest that the link between strong and enduring social relationships and costly collective acts is not uniquely human.

## Methods
**Data collection.** We conducted our study on adult (>12 years) male and female chimpanzees (*Pan troglodytes verus*) of three different communities (i.e., North, South and East) at the Taï National Park, Côte d'Ivoire (5°45′N, 7°07′W). Christophe Boesch started habituation efforts with North group in 1979 ([61] North: 1979; South: 1993; East: 2000; see Fig. 4 for home range estimations of the three groups), and established systematic observations and the collection of demographic data. Once the chimpanzees were habituated to researcher presence, trained observers began nest to nest focal-follow data collection[62] of adult individuals from each community on a daily basis (North: 1990–present; South: 1999–present; East: 2007–present). Focal-follow data included documentation of all changes in activity, social interactions (e.g., grooming) and vocalizations involving the focal individual[63]. Furthermore, due to the dynamic fission–fusion social system of chimpanzees[61,64] we continuously recorded of all changes in party size and composition.

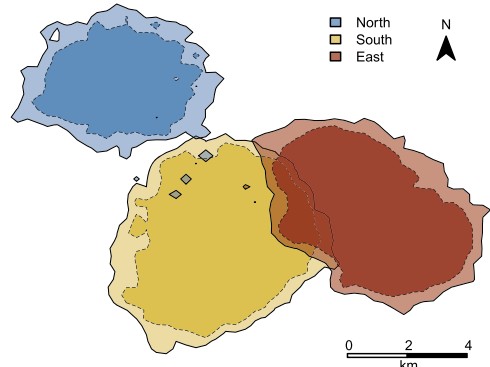

**Fig. 4 Home range estimations of North, South, and East groups.** Presented are the 95% (dashed lines) and 99% (solid lines) kernel home range utilization areas of North group (blue), South group (yellow), and East group (red). The 95% and 99% kernel home range sizes were 19 and 30 km² in North, 38 and 49 km² in South, and 32 and 43 km² in East, respectively. The home range overlap at 99% Kernel was 1.3 km² between North and South, and 15 km² between South and East. There was no home range overlap between North and East.

The study was done in compliance with the ethics policy of the Max Planck Society and was approved by the Ministries of Research and Environment of Côte d'Ivoire, and Office Ivoirien des Parcs et Réserves.

**Intergroup encounters.** Chimpanzee intergroup relations include patrols around border areas and into the territories of neighbours, and hostile and aggressive interactions with neighbours upon encounter[37,44,45]. Both types of territorial activities are associated with potential costs and unpredictability of outcome[23]. However, here, we focused on intergroup interactions that resulted in vocal or physical encounters with neighbours (hereafter intergroup encounters), as those pose a greater direct risk to participating individuals.

To examine what influences males and females' participation propensity in intergroup encounters we used detailed all occurrence data on intergroup encounters collected between the years 1997–2018. If an intergroup encounter was recorded, observers interrupted the focal follow data collection to document all occurrence data on participation, vocalizations and aggressions observed. In our analyses, we only included intergroup encounters in which at least one within-group member actively approached and engaged in the conflict and marked those that did so as participants. Thus, cases of intergroup encounters in which all party members showed no reaction to vocalizations by neighbours and/or retreated into their core area after detecting neighbours auditory and/or visually (e.g. ambush) were excluded, as no voluntary participation occurred[22]. Non-participation also meant that we could not evaluate how relationships with participants may contribute to participation, or what causes some community members to participate in the encounter but not others. By limiting our data to intergroup encounters involving an active approach or vocal response towards a neighbouring group, we can reliably investigate influences of inter-individual variation on participation.

For a subset of the intergroup encounter data with an active response (collected between the years 2013–2018 and comprising almost half of the entire dataset) we had accompanying information on whether or not intergroup encounters involved a border patrol. Border patrols are often initiated while in core areas and include cohesive movements of in-group members toward and beyond the borders of their territory[23], during which participation is considered voluntary[22]. Overall, more than 92% of active intergroup encounters followed a border patrol behaviour, thus, this approach sufficiently accounts for participation decisions. We differentiated between two types of active intergroup encounters both involving an active approach towards the out-group but that differ in the degree of contact, either (a) only vocal (vocal encounters) or (b) also visual and/or physical (contact encounters).

**Relatedness.** Maternal kinship and paternity for individuals of all ages of the different groups was established by means of pedigree and genetic data extracted from faecal samples[65] in the genetic lab of Linda Vigilant at the Max Planck Institute for Evolutionary Anthropology. The current Taï genetic dataset includes 259 individuals with an average of 83% complete genotypes at 19 autosomal loci. In brief, we extracted faecal samples (~100 mg) using either the QIAamp DNA stool kit (Qiagen) or the GeneMATRIX Stool DNA Purification Kit (Roboklon) following manufacturer instructions. We first simultaneously amplified aliquots at all 19 loci and subsequently reamplified using fluorescently labelled primers as detailed in ref. [66]. To confirm individual identities and assign paternities we

compared the resultant genotypes using the 'identity analysis' or the 'parentage analysis' functions of CERVUS, respectively, with confidence assessments of 80% and 95%. Potential sires were excluded by two or more mismatches such that each paternity assignment was of high likelihood.

**Dominance ranks**. We determined within-group dominance relationships and their changes over time by applying a likelihood-based adaptation of the Elo rating approach[67–69]. We used submissive uni-directional pant grunt vocalizations to estimate the dominance hierarchy within each of the sexes and groups separately (males: 9064, 10,011 and 4704 pant grunt vocalizations in North, South and East, respectively; females: 966, 1302 and 207 pant grunt vocalizations in North, South and East, respectively), standardized to a range from 0 to 1.

**Social bonds**. We used focal follow data collection of grooming behaviour (grooming bouts: North—14,599, South—18,948, East—9,040) to assess dyadic relationship quality following Samuni et al.[16]. In brief, we implemented the dynamic dyadic sociality index method[16,69,70], which provided continuous directed daily measures for grooming for each combination of within-group adult individuals. Using this method, dyads achieve high values by investing in grooming with one another more than others but maintain stable high values only if the investment provided was regular and consistent. We then used the directed grooming scores to evaluate preferences of grooming partners separately for same and opposite sex dyads. If both individuals of a dyad were each other's top preferred grooming partners, this dyad received a score of 1. Conversely, when the top grooming preference was one sided or entirely absent the dyad received a score of 0. As such dyads who scored 1 were those with mutual and stable grooming relationships. Using this method in the same population, we have previously shown that social bond partners are more likely to engage in the cooperative behaviour of food sharing than non-social bond partners[16].

**Dyadic association**. Chimpanzees live in a fission–fusion social system with transient and dynamic associations[61,64]. Association tendencies of males and females in Taï show intraindividual repeatability across days and years[71], indicating temporally stable phenotypes of gregariousness in this population which may influence participation likelihood. Thus, to evaluate spatial proximity, we used party association data collected during the year prior to the intergroup encounter date to construct matrices of dyadic association values of all adult individuals in a community. The time frame of one year prior to the encounter provides sufficient data to construct reliable association matrices, but as well likely the best representation of current social conditions (e.g., demography, group size, or female reproductive status) as the encounter which might influence association patterns. The dyadic association values were based on the duration two individuals were observed in the same party as a function of the total duration each of them was observed in total. For each intergroup encounter event we calculated the average dyadic association values with all other adult individuals that participated in the encounter. Higher average association values represent individuals that more frequently associated with encounter participants in comparison to lower association values. Thus, we could account for association degree with encounter participants in shaping participation decisions (see "Statistical analysis").

**Fruit availability index**. Chimpanzees use their home range according to the distribution of food sources. Thus, to account for the potential effect of food availability on participation likelihood, we calculated a monthly fruit availability index following a standard index for Taï chimpanzees[72]. We compiled the index based on the absence/presence of mature fruits, and the density and mean basal area of tree species. Phenology data was collected within the home-range of each chimpanzee group, thus, reflecting local variation in fruit productivity.

**Statistical analysis**. We fitted generalized linear mixed models (GLMM[73]) with binomial error structure and logit link function, to investigate variables affecting the likelihood to participate in intergroup encounters for adults. We evaluated 'participation decisions' by determining for each intergroup encounter all living adult community members and whether they participated or not (0/1). By including all adult individuals in the community as potential participants, we attempt to capture the patterns that predict the likelihood of some group members to engage in intergroup encounters but not others. This definition follows a previous study in chimpanzees[22]. As different factors may affect participation likelihood of the different sexes (e.g., reproductive state of females) we fitted separate models for the two sexes ('male participation model' and 'female participation model').

To test whether more cohesive engagement of group members (i.e., number of participants) affected participation in intergroup encounters via 'strength in numbers'[37,39,41], we included the number of other adult male and other adult female participants (not including the individual itself) as two separated test predictors. Furthermore, to test whether predictability of coalitionary support and interaction exchange is a mechanism influencing intergroup encounter participation, we investigated the effects of social preference and relatedness on participation likelihood. Hence, we included the presence or absence of adult maternal kin and social bond partners (preferred grooming partner) in the

encounter as additional test predictors. By testing the two hypotheses (i.e., number or preference of participants) in a single analysis we can evaluate their independent contribution to the response.

To assure that participation patterns are not an artefact of association patterns with other participants and in order to account for geographic decay (i.e. decrease in social ties with increased spatial distance between individuals), we included the average of the dyadic association indices with all participants (i.e., spatial proximity) as a control predictor in both models.

We controlled for an additional set of predictors that may affect participation likelihood. Individuals may gain from intergroup encounter participation, through inclusive fitness, if they have many relatives in the community. Thus, in both models we included the number of living maternal kin group members of all ages (for males—mother, maternal sibling, and maternal sister's offspring; for females—offspring and daughter's offspring) as a control predictor. We also included males' paternity success assessed by the number of living offspring in the community at the time of the encounter, as a control predictor for males. These potential benefits follow previous studies[22,25,49].

Males are the philopatric sex in chimpanzees with high male–male competition over mates, and over time successful intergroup encounters may lead to territory expansion[35] which may attract mating partners. If intergroup encounter participation facilitates future access to mates, increased reproductive opportunities could benefit males, but as well their mothers (through inclusive fitness). Thus, mothers may participate in intergroup encounters to foster future reproductive opportunities for their sons. Standard physical and behavioural criteria suggests that late juvenility/early adolescent life stages of male chimpanzees start at the age of 8[64,74], confirmed by physiological measures of the onset of puberty[75]. Thus, to investigate whether mothers are more likely to participate in intergroup encounters if they have sons approaching reproductive age, we included the presence of sons above the age of 8 in the group as a control predictor (independent son yes/no) for females.

Energetic constraints also influence intergroup encounter participation across primate species, with the prediction that the costs are higher in older and lower ranking individuals (reduced physical condition)[22,24–26]. Therefore, in both models, we included the squared term of age and dominance rank of individuals as potential costs indicating individuals' physical condition. In addition, we accounted for other potential energetic costs that may affect participation likelihood of females. Since pregnancy and early lactation are energetically demanding (considerably more in the first two years of lactation[76]) we controlled for the number of days into pregnancy and the presence of an offspring under 2 year of age as potential costs[25,26]. We estimated the onset of gestation by subtracting 228 days (average gestation length[77]) from parturition date. The year and month of birth was known for all offspring, and we assigned the 15th day of the month in cases where the day of birth was unknown. We as well accounted for females' reproductive status (i.e., maximal tumescence yes/no) Finally, we accounted for fluctuations in fruit availability driving participation of males and females[25]. In both models we controlled for community membership (North, South or East) and included the type of the encounter, whether it was strictly vocal or involved visual or physical contact, the latter we assume to be riskier. We also added the log-transformed number of potential participants (i.e., adult community size) as an offset term. Adding the number of potential participants as an offset term allowed us to account for differences in participation likelihood due to overall community size (e.g., general participation might be higher/lower in smaller/larger communities[22,27]).

Due to high correlation between the number of living maternal kin of all ages in the community and the presence of adult maternal kin in the encounter for males (Pearson's $R = 0.67$) we could not include both in the 'male participation' model. Thus, for the males we fitted two models that were identical in all terms except of including either the number of maternal kin or the presence of adult kin as predictors. In the "Results" section we present the results of the model with the lower AIC score, as this model performs better at explaining the response[78]. Nonetheless, we provide full information on the results of the other model in Table 3. This was not a problem in the female model as the number of living maternal kin in the community was mostly influenced by the presence of non-adult offspring.

To avoid pseudo-replication and account for non-independent sampling of certain individuals or intergroup encounters disproportionally affecting participation likelihood, we included the identities of encounters and potential participants as random effects. To keep type I error rate at the nominal 5%, we included the maximal random slope structure[79,80]. We thus included random slopes for age (linear and squared), dominance rank, number of maternal kin, number of offspring for males, and average dyadic associations within both random effects. In addition, we included the number of male and female participants, fruit availability and number of days into gestation within the random effect of potential participant. See Supplementary Code 1 for the complete specification of the full and null models. Our datasets included 343 intergroup encounter events with 75 potential female and 36 potential male participants, of three communities, resulting in 3,799 participation decisions for females and 1,377 participation decisions for males.

We processed the data and fitted all models in R (version 4.0.2[81]), using the function *lmer* of the R package 'lme4'[82]. Before fitting the models, we z-transformed all of the covariates to a mean of zero and a standard deviation of

**Table 3 Second 'Male participation' GLMM testing the effect of strength in numbers and predictability of coalitionary support on participation likelihood (Here, the number of living maternal kin is included instead of adult maternal kin presence in the encounter. $N = 342$ encounters, 36 male subjects).**

| Term | Reference category | Estimate | SE | 95% CI |
|---|---|---|---|---|
| Intercept | | −1.100 | 0.280 | −1.663, −0.415 |
| Number kin[a] | | 0.081 | 0.166 | −0.243, 0.435 |
| Bond partner (present) | Absent | 0.737 | 0.240 | **0.317, 1.250** |
| Number females[b] | | 0.587 | 0.117 | **0.356, 0.830** |
| Number males[c] | | 0.482 | 0.143 | **0.218, 0.777** |
| Dyadic association[d] | | 0.464 | 0.137 | **0.182, 0.752** |
| Number offspring[e] | | −0.117 | 0.155 | −0.427, 0.192 |
| Dominance rank[f] | | 0.236 | 0.133 | −0.021, 0.503 |
| Age[g] | | −0.383 | 0.261 | −0.920, 0.162 |
| Age[g]—squared | | 0.121 | 0.197 | −0.296, 0.534 |
| Type (vocal) | Contact | −0.122 | 0.191 | −0.522, 0.235 |
| Food availability[h] | | −0.010 | 0.093 | −0.204, 0.176 |
| South group | East group | −0.245 | 0.327 | −0.903, 0.475 |
| North group | East group | 1.038 | 0.524 | **0.017, 2.195** |

In bold appear the CIs that do not overlap 0.
The coded levels are indicated in parenthesis.
[a-f]z-transformed, mean ± SD of the original variables: [a]0.59 ± 1.05, [b]5.28 ± 3.53, [c]3.04 ± 1.29, [d]0.51 ± 0.18, [e]2.35 ± 2.38, [f]0.69 ± 0.26 (range 0-1 with 1 being the highest social rank), [g]21.31 ± 8.91, [h]1.81 ± 1.37.

one[83] and presented the original distribution of covariates in the table legends. We applied the function *vif* of the R package 'car'[84], to a standard linear model (lacking the random effects and slopes) to derive variance inflation factors (VIF), which revealed no collinearity issues (largest VIF: 'male participation model' = 2.11; 'female participation model' = 1.93[85]). Using a likelihood ratio test we evaluated model significance by comparing the fit of the full models with those of a respective null model lacking only the four test predictors[86]. To obtain individual *p*-values for all fixed effects we compared the full model with a series of models in which each fixed effect was systematically dropped one at a time, using the *drop1* function in R[79]. Model stability, assessed by comparing the estimates of the full model with those derived from a series of models excluding the different levels of the two random effects (identities of potential participants and encounters) one after the other, revealed no influential identities. We report the confidence intervals which were derived by means of parametric bootstraps with the function *bootMer* of the package 'lme4'. We calculated models' effect sizes ($R^2$) using the function *r. squaredGLMM* from the R package 'MuMIn', and report the variance explained by the fixed (marginal-$R^2$) and fixed and random (conditional-$R^2$) effects[87].

**Reporting summary**. Further information on research design is available in the Nature Research Reporting Summary linked to this article.

## Data availability
The data reported in this paper are available as Supplementary Data 1. Source data are provided with this paper.

## Code availability
The R code used to fit the models in this paper is available as Supplementary Code 1.

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

## Acknowledgements
We are grateful to the Ministère de l'Enseignement Supérieur et de la Recherche Scientifique and the Ministère de Eaux et Fôrests in Côte d'Ivoire, and the Office Ivoirien des Parcs et Réserves for permitting the study. We are grateful to Christophe Boesch for his dedicated effort over 40 years of research with the Taï Chimpanzee Project. We also thank the Centre Suisse de Recherches Scientifiques en Côte d'Ivoire and the Taï Chimpanzee Project staff members for their support, Linda Vigilant for conducting the paternity analysis, and to Erin Wessling and Roger Mundry for helpful discussions. Core funding for the Taï Chimpanzee Project was provided by the Max Planck Society since 1997. This work was also funded by the European Research Council under the European Union's Horizon 2020 research and innovation programme (grant agreement number: 679787) awarded to C.C.

## Author contributions
L.S., C.C., and R.M.W. conceptualized the study; L.S. designed the study; L.S. and R.M.W. managed data collection; L.S. compiled data for this study; L.S. collected data in the field; C.C. and R.M.W. provided infrastructure and logistical support for data collection; L.S. analysed the data; L.S. drafted the paper with feedback from all co-authors who also approved the final draft.

## Funding

## Competing interests
The authors declare no competing interests.
