## [Peer Review File · Nature Communications]

Reviewers' Comments:

Reviewer #1:

Remarks to the Author:

This paper examines the large and detailed dataset on intergroup encounters among the Tai chimpanzees to ask key questions related to the evolution of group-level cooperation: to what extent does participation depend on numerical advantage, the presence of kin, and the presence of non-kin social partners? The results provide compelling support for the importance of all three factors. Additionally, the study usefully examines both male and female participation in encounters, finding that individuals of both sexes participate, though males do so to a greater extent than females.

I have only a few minor comments:

L 149: "Whereas the number of living maternal kin in the group had no clear significant effect" "Group" is an ambiguous term in chimpanzee studies. Do you mean the whole community, or the party under observation during the encounter? I suggest using either "community" or "group," as appropriate, to improve clarity.

L. 151-152: "In females, the total number of living maternal kin in the group had a significant negative effect"

This is an odd finding! However, as noted later (L. 217), "the adult maternal kin present in the intergroup encounter were mainly their sons." This suggests that this odd result is a consequence of (1) including sons in the model and (2) the very small number of other maternal kin for females. I would therefore recommend excluding this covariate from the model for females, since it is (1) mostly redundant with sons and (2) likely leading to a spurious result given the small number (N=2) of other maternal kin for females.

L. 224-225: "Contrary to expectations of risk assessment, we found that contact intergroup encounters were positively associated with female participation"

Is this result really contrary to "expectations of risk assessment?" Risk assessment should include not only the risks of being injured, but also the value of the resource at stake. If females are fighting to defend their infants, or fighting to ensure their own safety, then participation may, on balance, have greater expected payoff than relying on the efforts of others.

See, for example, the playback studies by Packer and Pusey's team in Serengeti and Ngorongoro Crater. Females increased participation in response to playbacks when the value of the resource at stake was higher: in the crowded habitat of Ngorongoro, and when cubs were present — and particularly when they have many allies present to reduce the risks of participation. Males almost always responded strongly to simulated intruders, presumably because their lifetime reproductive success is always on the line when facing male intruders.

Note also that (as you point out earlier in the paper) the costs of participation decrease with increasing numerical advantage. So it may not be so risky for females to participate in contact encounters, provided many other allies are also present. (Most killings in chimpanzees occur when individuals are greatly outnumbered, rather than when parties with many individuals on both sides meet.)

Reviewer #2:

Remarks to the Author:

With great interest I read this MS investigating the social (in particular), and ecological variables associated with individual participation in the intergroup encounters among three neighboring communities of western chimpanzee. Collecting sufficiently detailed data on such interactions between

groups of wild animals is no small feat, and I commend the authors on a tremendous effort. It is also apparent that a lot of time and effort went into the preparation and analyses of these data, for which the authors also deserve credit.

Much to my surprise, however, not a single reference was made to the large and ever growing (see e.g. a recent online special issue of the International Journal of Primatology) body of literature on this very same topic in other non-human primates (or other animals, for that matter), an omission that struck me as a missed opportunity. In my opinion, the authors do their own work a disfavor by exclusively drawing parallels to (the very few) empirical and (not entirely uncontroversial) theoretical studies on large scale cooperation and intergroup interactions in humans, instead of also trying to place their findings in a broader, more comparative framework. Such comparisons to other animal taxa would be particularly insightful, given that some of the reported findings seem to contrast with fairly consistent patterns across the primate order (specifically, the absence of an effect of age, dominance and food availability on an individual's decision to participate in intergroup encounters are very interesting in this respect).

In addition, and regrettably, I found the wording often rather convoluted, to the extent even that I sometimes struggled to follow/recognize the logic of the argumentation. I further felt that many decisions on which data to use, and how to score key variables in the analyses, require more explanation/justification than is currently present in the text, to be able to properly assess the significance and implications of the reported results.

Because of these reservations (and more specific ones, I elaborate on below), I cannot recommend this manuscript for publication in Nature Communications. I encourage the authors to critically rethink both the narrative pitch and most appropriate outlet (space seems to be a limiting factor here!) for their valuable, timely and interesting study.

Further comments/suggestions:

Line 14-16:

An information-packed sentence that I would love see explained in more detail in the introduction. Specifically: what are these social benefits that extend beyond the dyad in humans, how do they promote our unique capacity for large scale cooperation, and why do you suspect these could -as you seem to imply: uniquely- be shared by chimpanzees?

Line 22-23:

Can the number of participants and their identity really be equated to "properties of a social network that facilitate group-level cooperation", or might you also, more mundanely perhaps, call them "social variables that affect individual-level decisions on participation"?

Line 32:

In my understanding of this literature, a key-property of human social networks thought to promote large-scale cooperation, is their multi-level nature (in line with this interpretation, you quote Dyble et al. 2016, a couple of lines down). I struggle to follow then, why you feel chimpanzees (that do not live in multi-level societies) make a particularly compelling comparative group in this regard to understand human social evolution.

Line 26-27

The evolutionary significance, existence and direction of this causal relation as suggested by these authors, is hotly debated, and empirically poorly supported (see e.g. several chapters in the edited volume by Fry 2013).

Line 44-48:

So again -apologies!-, I fail to see how you are looking at higher level structures of a social network, and not simply at traits of dyadic relationships that an individual may (appear to) use in its cooperative decision making process.

Line 50-51:

"the distribution of the number of social ties" or more plainly "the number of in-group participants"? What about collective action problems, which are thought to be common in the intergroup relations of primates, including chimpanzees (e.g. Kitchen & Beehner 2007, Willems et al. 2013, Langergraber et al. 2017)

Line 55-57:

Kin selection and reciprocity evaluated simultaneously! Nice.

Line 77-78:

I was surprised to see you limit your analyses to "initiated" or "active approach" scenarios by the focal groups, and am not sure I follow the justification for this decision (lines 295 -297). In fact, I would argue it is crucial to also consider scenarios in which individuals decide NOT to participate in an intergroup encounter to be able to get the complete picture.

It was also unclear to me how you defined "the group": was this e.g. a foraging party, a boundary patrol, or always the entire community, and was participation defined as simply being present (which I believe is the criterion you used?), or as active behavior (e.g. vocalizing or threatening/aggressively approaching a member of the opposing group, as is more commonly the case in studies on intergroup encounters). These are critical elements required to be able to interpret your findings, yet are hard to extract from the Methods section.

Line 85:

"social distribution and preference"= "number of in-group participants and their identity (in terms of relatedness/social bond)"?

Line 87:

"social bond" or "preference" vs. "geographic decay": it wasn't clear to me at all (until I reached lines 326 - 349) that you're basically using these terms to refer to two dyadic sociality indices, one based on grooming, the other on association. I think you could considerably help your readership by briefly defining these terms earlier on. Also, why did you transform the former to a binary predictor, and the later to an average across all current party members (is this biologically meaningful!)?

Lines 89-98:

Here (but also previously) I feel you should really refer to the extensive literature on intergroup encounters in non-human animals/primates, and formulate more explicit expectations based on this previous work.

Lines 115-116:

From the degrees of freedom alone, it is not clear here what your null-models actually were.

Line 117 -130

Some readers, including myself, might be interested in learning about how much of the total variance your models actually accounted for. Also, were any biologically plausible interactions considered (e.g. food availability might matter, but only if close relatives and/or social partners are present)? If so, which, and why (not)?

Line 173-175:

These negative results are really interesting! Why would this be the case in chimpanzees but not in many other non-human primates?

Line 245-246:

Again, see edited volume by Fry 2013 (War, Peace and Human Nature)

Line 249-253:

I feel here the discussion really wanders off too far from what your results may plausibly imply...

We thank both reviewers for their valuable and constructive comments and for dedicating their time to evaluate our manuscript during this unpredictable time. We believe their comments have helped to improve the coherence and clarity of our manuscript. We have addressed all comments either in the manuscript and/or as a comment reply and provide point-by-point responses to their comments (in red) in the following lines.

Reviewer #1 (Remarks to the Author):

This paper examines the large and detailed dataset on intergroup encounters among the Tai chimpanzees to ask key questions related to the evolution of group-level cooperation: to what extent does participation depend on numerical advantage, the presence of kin, and the presence of non-kin social partners? The results provide compelling support for the importance of all three factors. Additionally, the study usefully examines both male and female participation in encounters, finding that individuals of both sexes participate, though males do so to a greater extent than females.

We are grateful for the reviewer for their comments on our manuscript.

I have only a few minor comments:

L 149: "Whereas the number of living maternal kin in the group had no clear significant effect" "Group" is an ambiguous term in chimpanzee studies. Do you mean the whole community, or the party under observation during the encounter? I suggest using either "community" or "group," as appropriate, to improve clarity.

We agree with the reviewer that "group" might be an ambiguous term here, and potentially in other text lines. In this sentence, and throughout the paper, we have used the term "group" to refer to the entire community. To avoid further ambiguity, we are now using the term "community" instead, throughout.

L. 151-152: "In females, the total number of living maternal kin in the group had a significant negative effect"

This is an odd finding! However, as noted later (L. 217), "the adult maternal kin present in the intergroup encounter were mainly their sons." This suggests that this odd result is a consequence of (1) including sons in the model and (2) the very small number of other maternal kin for females. I would therefore recommend excluding this covariate from the model for females, since it is (1) mostly redundant with sons and (2) likely leading to a spurious result given the small number (N=2) of other maternal kin for females.

This is an important point to clarify which we now address in the discussion and method sections.

To summarize, although the two variables the reviewer mentioned appear similar, they vary greatly and represent largely different assessments of relatedness. The total number of living maternal kin in the community included individuals of all ages. For our study population this variable was predominantly influenced by the number of non-adult offspring a female had (variable ranging between 0-5). In contrast, the second variable (i.e., presence of adult kin in the encounter) included individuals >12 years of age (adults), and only those that were present in the encounter. Across the three communities we find females that have several offspring under the age of 12 but none above that age, and vice versa. Thus, these two variables are not expected to correlate considerably. Nonetheless, prior to fitting the models we tested for collinearity between all model predictors. This procedure revealed no collinearity issues for

these two predictors in the female model (VIF: adult kin in party = 1.72, number of living maternal kin = 1.93), meaning that model assumption of un-correlated predictors was not violated.

Conversely, for male participants, we found a strong correlation between the number of living maternal kin (of all ages) in the community and the presence of adult maternal kin in the encounter (discussed in lines 459-461). This is because, by definition, their offspring are not categorized as maternal kin but modelled via a third predictor – number of living offspring. Due to the correlation between these two predictors for males, and as the reviewer suggested, we previously fitted two separated models (described in lines 461-465) and present the results of both models in the main text and supplementary materials.

Given that the total number of living maternal kin mostly encompasses non-adult individuals, and the lack of collinearity with the presence/absence of kin for females, we believe it is valuable to include both in the same analysis. Including both in the same analysis allows us to distinguish whether it is the degree of relatedness with other community members that influences participation, or the coordination with related group members during the encounter.

Nonetheless, as a post-hoc investigation and to assure that model results are truly not influenced by the jointly inclusion of these two parameters, we fitted two reduced female models each excluding one of these predictors or the other. This post-hoc exploration revealed that our results regarding the effect of both kin predictors are robust, as we still find negative effect of the number of living maternal kin in the community (estimate \pm SE = -0.23 ± 0.12 ; $P = 0.021$), and a positive effect of the presence of adult maternal kin in the encounter (estimate \pm SE = 1.44 ± 0.28 ; $P < 0.001$). The other model results as well remained the same.

We suspect that the negative significant effect of the number of living maternal kin in the community likely results from the fact that this variable is predominantly influenced by the number of non-adult offspring. As young chimpanzee still regularly associate with their mother until adolescence periods, females with more non-adult offspring may have higher potential participation costs due to risk to their offspring. We have added further elaboration on this point into the discussion (please see lines 227-230).

L. 224-225: “Contrary to expectations of risk assessment, we found that contact intergroup encounters were positively associated with female participation”

Is this result really contrary to “expectations of risk assessment?” Risk assessment should include not only the risks of being injured, but also the value of the resource at stake. If females are fighting to defend their infants, or fighting to ensure their own safety, then participation may, on balance, have greater expected payoff than relying on the efforts of others.

See, for example, the playback studies by Packer and Pusey’s team in Serengeti and Ngorongoro Crater. Females increased participation in response to playbacks when the value of the resource at stake was higher: in the crowded habitat of Ngorongoro, and when cubs were present — and particularly when they have many allies present to reduce the risks of participation. Males almost always responded strongly to simulated intruders, presumably because their lifetime reproductive success is always on the line when facing male intruders.

Note also that (as you point out earlier in the paper) the costs of participation decrease with

increasing numerical advantage. So it may not be so risky for females to participate in contact encounters, provided many other allies are also present. (Most killings in chimpanzees occur when individuals are greatly outnumbered, rather than when parties with many individuals on both sides meet.)

We fully agree with the reviewer and have edited this paragraph accordingly. Specifically, we emphasize that although contact intergroup encounters likely involve higher risk of injury, such risk can be reduced with an overall increase in participation. We also use the aforementioned lion studies to contextualize the 'strength in number hypothesis' and inform our predictions and discussion. Further, we discuss that our results are similar to a study in mountain gorillas showing that more aggressive intergroup encounters involved a higher proportion of participating individuals (please see lines 243-252).

Reviewer #2 (Remarks to the Author):

With great interest I read this MS investigating the social (in particular), and ecological variables associated with individual participation in the intergroup encounters among three neighboring communities of western chimpanzee. Collecting sufficiently detailed data on such interactions between groups of wild animals is no small feat, and I commend the authors on a tremendous effort. It is also apparent that a lot of time and effort went into the preparation and analyses of these data, for which the authors also deserve credit.

We are grateful to the reviewer for their positive feedback.

Much to my surprise, however, not a single reference was made to the large and ever growing (see e.g. a recent online special issue of the International Journal of Primatology) body of literature on this very same topic in other non-human primates (or other animals, for that matter), an omission that struck me as a missed opportunity. In my opinion, the authors do their own work a disfavor by exclusively drawing parallels to (the very few) empirical and (not entirely uncontroversial) theoretical studies on large scale cooperation and intergroup interactions in humans, instead of also trying to place their findings in a broader, more comparative framework. Such comparisons to other animal taxa would be particularly insightful, given that some of the reported findings seem to contrast with fairly consistent patterns across the primate order (specifically, the absence of an effect of age, dominance and food availability on an individual's decision to participate in intergroup encounters are very interesting in this respect).

The reviewer has voiced their concern that the background, predictions and discussion of our study are lacking, in the sense that we do not reference to the existing literature on intergroup encounter participation in non-human animals. We addressed this concern by expanding the literature used to inform our predictions, inclusion of control predictors and when contextualizing results of significant predictors.

In sum, in this study we focus on the link between various social aspects and intergroup encounter participation. We aim at investigating a question that to our knowledge remains largely unexplored in non-human animals, whether differentiated social relationships between kin and non-kin promote collective action. For this reason, we focus our predictions and discussion around the tested social parameters (i.e., number of other participants, and presence of adult maternal kin or bond partners), and have incorporated additional reference to existing literature when relevant to our hypotheses, predictions and discussion.

Additionally, for a reliable assessment of social parameters, we controlled for a set of variables that are frequently tested in the context of intergroup relations (i.e., dominance rank, age, food availability). As these are control predictors, we neither provide explicit predictions for their effects nor discuss their results in length. Instead, we have now incorporated additional literature to justify the inclusion of our set of control predictors and emphasize that controlling for these variables is essential for revealing whether our test predictors independently affect intergroup encounter participation decisions.

Given the large amount of studies investigating the effects of some of our control predictors but the rarity of studies investigating the effect of the social parameters, we believe that focusing our paper on social aspects likely has a larger contribution to the field of intergroup relations, social relationships and cooperation. We hope that the reviewer appreciates this perspective.

In addition, and regrettably, I found the wording often rather convoluted, to the extent even that I sometimes struggled to follow/recognize the logic of the argumentation. I further felt that many decisions on which data to use, and how to score key variables in the analyses, require more explanation/justification than is currently present in the text, to be able to properly assess the significance and implications of the reported results.

Per the reviewer's comment, we now include more precisely defined key terms and further clarify explanations on the choice of data used. Please see our specific responses to all comments, below and throughout the manuscript text.

Because of these reservations (and more specific ones, I elaborate on below), I cannot recommend this manuscript for publication in Nature Communications. I encourage the authors to critically rethink both the narrative pitch and most appropriate outlet (space seems to be a limiting factor here!) for their valuable, timely and interesting study.

Further comments/suggestions:

Line 14-16:

An information-packed sentence that I would love see explained in more detail in the introduction. Specifically: what are these social benefits that extend beyond the dyad in humans, how do they promote our unique capacity for large scale cooperation, and why do you suspect these could -as you seem to imply: uniquely- be shared by chimpanzees?

The first sentence of the abstract is a brief summary of the first paragraph of the introduction. In sum, humans maintain a wide range of social relationships with varying degree of reciprocal exchange. This wide range of differentiated relationships is known promote cooperation on a dyadic level (e.g., exchange of resources between partners) but as well on a larger scale (our meaning with "beyond the dyad"), for example when occurring amongst groups (large-scale cooperation) or when several individuals are acting together (collective action; please see lines 31-35). We have edited the sentence to clarify our meaning (lines 14-16).

We did not intend to imply that large-scale cooperation is a capacity shared with chimpanzees and have now further clarified this both in the abstract and the introduction. Rather, we draw similarities in human and non-human animals' capacity to maintain differentiated social relationships that potentially promote cooperation. As non-human animals are also capable of collective action, a question remains of the role of differentiated social relationships in promoting

collective action in non-human species. Specifically, we note: “Similarly, non-human animals maintain differentiated social relationships that promote dyadic cooperative exchange, but their link to cooperative collective action is little known.” (lines 16-18).

Regarding examples of benefits. In the first sentence of the introduction we detail some of the benefits of maintaining social relationships, such as improved fitness and wellbeing. Later in the manuscript, we discuss key benefits that are potentially gained from collective cooperation, like support of human costly life history trajectories and demanding foraging niche (lines 284-286).

Line 22-23:

Can the number of participants and their identity really be equated to “properties of a social network that facilitate group-level cooperation”, or might you also, more mundanely perhaps, call them “social variables that affect individual-level decisions on participation”?

The reviewer is correct that we have not properly defined our intended definition of social networks in the abstract, thus the use of “properties of social network” here is unwarranted. As a response we have changed the abstract sentence: “Together, maintaining a range of differentiated social relationships facilitated group-level cooperation”

The term social network properties throughout the manuscript refers to the variety of differentiated social relationships an individual has in their group (see lines 31-33 for definition). Some examples for social network properties are the number of interacting members in the social network (i.e., distribution degree; here the number of participants), the degree of strength and reciprocity between members of the social network (here, social bond partners and adult maternal kin) , and even spatial proximity between members of the social network (here, association degree). Varying levels of dyadic interaction exchange is the basis of the social network. We have clarified throughout the text.

Line 32:

In my understanding of this literature, a key-property of human social networks thought to promote large-scale cooperation, is their multi-level nature (in line with this interpretation, you quote Dyble et al. 2016, a couple of lines down). I struggle to follow then, why you feel chimpanzees (that do not live in multi-level societies) make a particularly compelling comparative group in this regard to understand human social evolution.

We apologize for the lack of clarity which may have led to some misinterpretation.

Like us, the Dyble et al., (2016) study focused their investigation on within camp (or for us, within community) exchange of interactions. Their study investigated Agta and Mbendjele food sharing behavior between different households located in a camp, with camp sizes ranging between 41-62 people (equivalent of a chimpanzee community size). They defined households as the “occupants of a single dwelling” and those were characterized by high degree of relatedness. They did not investigate food sharing behavior between different camps, which is what you would expect if focusing on higher levels of the multi-level society (which we agree with the reviewer chimpanzees do not have).

As such, the fission-fusion social groups of chimpanzees can be seen as a camp, with some individuals that associate frequently (i.e., association affinity), or nearly permanently (i.e., kin)

resembling the degree of the “household” in the Dyble study. Chimpanzee communities also have un-related individuals that positively interact in a preferred manner (e.g., social bond partners), resembling the “household clusters” in the Dyble study.

Dyble et al., suggested that the structuring of the Agta and Mbendjele camps into several clusters of households that more preferably exchange commodities, represent a multilevel social organization that supports human costly life history. People in the multilevel social organization (i.e., camp) do not interact equally with all other camp members, but instead maintain a differentiated range of relationships that sustains their needs.

Thus, the argument here is that it is not about which degrees or levels of the multi-level society that influence cooperation. The argument here is that within-community or within-camp clustering of relationships allows access to a range of cooperation capacities.

In that manner, chimpanzees and other group living social animals are good models to investigate the link between differentiated social relationships and cooperative capacities.

We have clarified in the abstract, introduction and discussion that humans’ capacity for large-scale cooperation is unparalleled (please see lines 14-16, 33-37, 286-289).

Line 26-27

The evolutionary significance, existence and direction of this causal relation as suggested by these authors, is hotly debated, and empirically poorly supported (see e.g. several chapters in the edited volume by Fry 2013).

[Reading the comment of the reviewer we assume that they referred to lines 43-44 and not 26-27 as stated] We agree with the reviewer that it is debated whether intergroup conflict was present throughout the evolution of our species or instead a more recent byproduct of possession and settlements. We have now removed this statement from the introduction and have edited the wordings in the discussion to address this (please see lines 277-278).

Line 44-48:

So again -apologies!-, I fail to see how you are looking at higher level structures of a social network, and not simply at traits of dyadic relationships that an individual may (appear to) use in its cooperative decision making process.

To provide more clarity we have edited the wordings in the sentence and define our meaning of social network. Specifically, social networks here do not refer to group-level properties, such as modularity or clustering (higher-level structures), but to the variety of relationships individuals within the social networks have (Apicella et al., 2012). These variety of relationships inform the properties of the social network.

Line 50-51:

“the distribution of the number of social ties” or more plainly “the number of in-group participants”?

The reviewer is correct and we have edited our wording to be more specific. We also reference to more studies to inform this prediction (see lines 51-55)

What about collective action problems, which are thought to be common in the intergroup relations of primates, including chimpanzees (e.g. Kitchen & Beehner 2007, Willems et al. 2013, Langergraber et al. 2017)

This is an important point and we now further clarified in the methods how we accounted for the potential negative effect of community size on collective action (please see detailed explanation at the end of this response).

Also, we would like to take this opportunity to discuss why we believe collective action problem should not affect our predictions regarding the 'strength in number hypothesis'.

A collective action problem may arise because the potential benefits of "successful" intergroup interactions (e.g., territory) are a public good that is accrued by the entire members of the community independent of their participation. Therefore, individual's incentives for intergroup encounter participation might be lower in larger groups because defection has a relatively smaller impact on success probability the larger the group becomes. Following this, the collective action problem refers to variation in sizes of communities with the prediction that larger communities are likely to suffer more from collective action problem relative to smaller communities.

Here, we do not investigate between-community variation, but focus on within-community, inter-individual, inter-encounter variation in participation. In this specific case, we investigated whether the number of other male and female participants increases participation likelihood of others.

The collective action problem and 'strength in numbers' are therefore independent mechanisms. For the first, the relevant parameter is the percentage of participating community members, for the latter the relevant parameter is the relative number of participants. For example, a large community can suffer from collective action problem in which on average only 30% of individuals join. Nonetheless, if during a certain encounter only 1 individual out of the 30% participates, then the chances for others to participate should be lower than if 10 participate. The same should be true if on average a higher percentage of the community participates, as could happen in smaller groups (in our study 86% of all male and 48% of all female community members participated on average).

Even if a certain community is smaller or larger in comparison to other communities, the participation incentives of community members should still be influenced by cost-to-benefit ratios, which in chimpanzees (and other animals) are largely regulated by numerical advantage. Therefore, the 'strength in numbers' hypothesis is relevant for communities of varying sizes.

The reviewer is correct that community size (and hence collective action problem) must be accounted for. This is because the total number of potential participants in the community (number of adult community members) has a direct influence on participation decisions. We included in all models an offset term of the adult community size. Through this offset term, we can test the effect of numerical participation while accounting for the community size and potential collective action problem (lines 455-458).

Line 55-57:

Kin selection and reciprocity evaluated simultaneously! Nice.

We appreciate the reviewer's comment.

Line 77-78:

I was surprised to see you limit your analyses to “initiated” or “active approach” scenario’s by the focal groups, and am not sure I follow the justification for this decision (lines 295 -297). In fact, I would argue it is crucial to also consider scenario’s in which individuals decide NOT to participate in an intergroup encounter to be able to get the complete picture.

We thank the reviewer for pointing this out and agree that “it is crucial to also consider scenarios in which individuals decide NOT to participate in an intergroup encounter to be able to get the complete picture”.

Specifically, our aim was indeed to model participation decisions at the individual level. Thus, following the reviewer’s comment we understand that our methodological description was unclear. We have edited the text to more clearly explain why our approach is essential for documenting reliable participation decisions in intergroup encounters (and not overall influences of when encounter occur or not).

To clarify our approach: In our analyses we focused on parameters that influence individual, and not community level, participation decisions. We wanted to ask, why do some individuals participate in a specific encounter, while others do not?

To do so, we needed to limit our data to intergroup encounters in which at least one community member actively approached, vocalized, and/or attacked the out-group, since otherwise no voluntary participation decisions were made. For example, we did not include intergroup encounter that involved an ambush by the out-group, because, as mentioned by Langergraber et al., (2017) “participation in intergroup aggression is not always entirely voluntary, because attacks by members of other groups demand immediate responses”.

Thus, limiting the data to intergroup encounters with an active response was a necessary step to assure that we capture voluntary participation decisions. Furthermore, if we were to include cases with no participation then we would not be able to assess some main predictors, like the degree of relatedness or social bond with encounter participants (as there will be no participants to assess these with).

We have clarified this in lines 315-328.

It was also unclear to me how you defined “the group”: was this e.g. a foraging party, a boundary patrol, or always the entire community, and was participation defined as simply being present (which I believe is the criterion you used?), or as active behavior (e.g. vocalizing or threatening/aggressively approaching a member of the opposing group, as is more commonly the case in studies on intergroup encounters). These are critical elements required to be able to interpret your findings, yet are hard to extract from the Methods section.

The reviewer is correct saying that it is important to provide clear definitions and that we should be more careful in our methodological description to assist comprehension of our approach, and we have edited the text accordingly.

In the paper we made the distinction between potential participants and actual participants.

We defined potential participants for every intergroup encounter as - all adult male and female living group members (following Langergraber et al., 2017; Mirville et al., 2018). Group here

refers to the entire community, and following a suggestion by Reviewer 1, we now use the less ambiguous term of “community” throughout the manuscript.

Participating individuals were defined as – only those individuals who were present in the party when the encounter started (due to the fission-fusion social structure of chimpanzee communities those are the ones we could observe) and also approached towards, vocalized, and/or attacked the out-group. Thus, for instance, if a certain individual was present in the party when neighbors were heard but this individual retreated instead of approaching (while other individuals approached), they were not marked as participants. If all party members retreated when neighbors were heard or ignored vocalizations from the out-group, then no participation has been made and such events were excluded.

We collected our intergroup encounter data as all occurrence data during focal follows. Once an intergroup encounter occurred (i.e., neighbors were seen or heard), observers ended their focal follow data collection and prioritized the documentation of the occurrence around the intergroup encounter. This meant that if certain party members approached and attacked the neighbors, the observers followed these individuals even if the focal individual was not part of the party that approached. Please see lines 315-328.

Line 85:

“social distribution and preference”= “number of in-group participants and their identity (in terms of relatedness/social bond)”?

We have now changed this section to be clearer about the exact test predictors used to test our two hypotheses. The text (lines 88-91) now reads “Following our predictions for the two hypotheses, we tested the effect of i) the number of other male and female in-group participants (‘strength in number hypothesis’), and ii) the presence of adult maternal kin or social bond partners (‘predictability of coalitionary support hypothesis’) on the likelihood to contribute to intergroup encounters in both sexes.”

Line 87:

“social bond” or “preference” vs. “geographic decay”: it wasn’t clear to me at all (until I reached lines 326 – 349) that you’re basically using these terms to refer to two dyadic sociality indices, one based on grooming, the other on association. I think you could considerably help your readership by briefly defining these terms earlier on. Also, why did you transform the former to a binary predictor, and the later to an average across all current party members (is this biologically meaningful?!)?

We now more clearly define all terms upon first mention.

Social relationships can be characterized in various ways, and often include assessments of the strength, mutuality, and stability of the relationship. Whereas the three components can be represented on a continuous scale, the distinguish of social bond partners is frequently done in a binary way (Crockford et al., 2013; Silk et al., 2009; Wittig et al., 2016), such that social bond partners are characterized by a strong and mutual interaction exchange that is stable across time. Furthermore, using the same approach for the assessment of social bonds, we previously shown that social bond partners in Tai are more likely to engage in another form of cooperative act, food sharing (Samuni et al., 2018). Thus, this likely represents a biologically meaningful

assessment of social bonds, at least for the Tai chimpanzee population. We have added this in lines 372-374.

For the association data, we wanted to account in the model for variation in participation that could be explained by association tendencies of individuals with the other participants. By accounting for associations, we could evaluate the effect of our test predictors independently of individuals' likelihood to be at the same place in the same time with participants. For that reason, each adult community member, whether they participated or not, received a value that was equal to the average of dyadic associations with all encounter participants. If a certain individual associate frequently with many of the encounter participants, then the average association value for this individual should be high, and vice versa. Our model results show that the average dyadic association value with encounter participants had a significant positive effect on participation likelihood. We have clarified this in the method section (lines 386-389).

Lines 89-98:

Here (but also previously) I feel you should really refer to the extensive literature on intergroup encounters in non-human animals/primates, and formulate more explicit expectations based on this previous work.

We appreciate the reviewer's comment and have expanded on the literature relevant for our hypotheses, predictions, and findings.

We would also like to take this opportunity to explain why we use some of the literature to only justify the inclusion of certain variables in the model (i.e., control predictors), but not to formulate their predictions.

In the paper we use a hypothesis testing approach via full-null model comparison. We test two sets of socially driven hypotheses (i.e., strength in numbers and predictability of coalitionary support). These hypotheses refer to specific "social" influences of intergroup encounter participation. For these hypotheses we provide a clear set of predictions supported by previously published literature that is now further expanded to include more studies and species. However, as the effects of the presence of a social bond partner or maternal kin on individuals' participation decisions in intergroup encounters is largely unexplored, the literature used to inform predictions of these variables was limited.

In order to reliably test for the effect of the test predictors, we must control for other variables known to influence intergroup encounter participation (control predictors). As the reviewer stated, there is extensive literature on factors influencing intergroup encounter participation such as food availability, presence of young infants, age, or dominance rank. These variables are included in our models as control predictors.

Since these are control predictors, we do not provide predictions for their effects or use previous literature to inform us on expectations of their effects. Instead, we used the previous literature to inform the different variables that one must account for when investigating intergroup encounter participation. As the control predictors were not the focus of our paper, we previously justified their inclusion by only referring to other chimpanzee studies. However, following the reviewer's suggestion, we refer to more studies that include more species when reporting on the variables controlled for in the models (please lines 95-111, 421-428, 440-453).

Lines 115-116:

From the degrees of freedom alone, it is not clear here what your null-models actually were.

Following the reviewer's comment, we have added information on the full-null model structures in both tables and the end of the introduction (in addition to the description in the methods section).

Line 117 -130

Some readers, including myself, might be interested in learning about how much of the total variance your models actually accounted for. Also, were any biologically plausible interactions considered (e.g. food availability might matter, but only if close relatives and/or social partners are present)? If so, which, and why (not)?

As the reviewer suggested we have now added the effect sizes (marginal and conditional R^2 , see lines 129-130) for both models. The marginal and conditional R^2 are a summary statistic quantifying the percentage of variance explained by the fixed effect (marginal) or the fixed and random effect (conditional) structure in the model (lines 492-295) (Nakagawa & Schielzeth, 2013).

Regarding the inclusion of biologically plausible interactions. Prior to fitting all the models, we detailed our predictions and the specific test and control predictors used. In the process, we also considered the inclusion of potential interactions. Important interactions that we considered were interactions including the sex of the individual with many of the other predictors, because theory suggests sex-differences in intergroup encounter participation. However, as we needed to account for different variables for each sex (e.g., reproductive state of females, offspring of males not being included as maternal kin) we instead fitted two different models, one for males and one for females. We did not identify any other biologically plausible interactions that should be included.

The reviewer suggested that food availability might influence participation only when a bond partner/kin is present. We would not expect this to be the case. We predicted that the presence of a bond partner/kin should increase participation because social bond partners/kin are more likely to provide social support that is necessary in potentially risky situations. As those types of relationships are characterized by stability over time, we predict that their effect on participation should not be influenced by the availability of food sources.

Line 173-175:

These negative results are really interesting! Why would this be the case in chimpanzees but not in many other non-human primates?

The negative results in this study are interesting but are no exception across studies and/or species. Many studies in non-human primates find a positive effect of dominance rank on encounter participation likelihood (female blue monkeys: (Cords, 2007); male sifakas: (Koch et al., 2016); male chimpanzees: (Langergraber et al., 2017); male gorillas: (Mirville et al., 2018)), initiation likelihood (female vervet monkeys: (Arseneau-Robar et al., 2017) or aggressiveness during encounters (baboons: (Kitchen et al., 2004); bonnet macaques: (Cooper et al., 2004); ring-tailed lemurs: (Nunn & Deaner, 2004); japanese macaques: (Majolo et al., 2005)).

However, other studies do not find an effect of dominance rank on participation likelihood (male chimpanzees: (Watts & Mitani, 2001; Wilson et al., 2001); ring-tailed lemurs: (Nunn & Deaner,

2004)) including a recent meta-analysis of 31 non-human primate species (Majolo et al., 2020), 2020). The same study found no significant effect of age, nor was age influential in chimpanzee participation in border patrols (Langergraber et al., 2017). For food availability, while several studies have shown that food availability positively influences encounter occurrence (Cooper et al., 2004; Lucchesi et al., 2020; Sakamaki et al., 2018), studies that specifically investigated whether food availability affects participation likelihood mostly found non-significant results (Koch et al., 2016; Mirville et al., 2018; Wilson et al., 2001). The same holds true for the other non-significant control predictors in our study.

Here, we did not attempt to interpret or discuss non-significant results for several reasons. First, interpreting non-significant results is challenging, as there is a large uncertainty associated with non-significance that cannot be reliably quantified. This is because both an absence of an effect or lack of power (of the specific predictor) can lead to non-significance and the two are difficult to distinguish. Also, it can be that the effect of variable A on a response Y is indirect through B (i.e., $A \rightarrow B \rightarrow Y$), and once B is also included in the model then the estimated effect of A on Y is close to zero. These independent issues make non-significant results difficult to interpret.

Further, interpreting non-significant results of parameters that have been investigated many times in the past in many different species, including chimpanzees, has relatively little scientific value in adding to the already large pre-existing literature on the topic. Instead, we focused our discussion on new and relatively rare investigations of “social” factors influencing participation decisions, which we believe has a greater contribution to current and future scientific discussions. We hope that the reviewer appreciates this perspective.

Line 245-246:

Again, see edited volume by Fry 2013 (War, Peace and Human Nature)

As mentioned in response to previous comment, we have edited this sentence. The edits instead discuss the proximate effects intergroup competition has on group social dynamics and cooperation in humans, for which there is multiple evidence (please see lines 277-278).

Line 249-253:

I feel here the discussion really wanders off too far from what your results may plausibly imply... On reflection, we agree with the reviewer that as it stands this section does not fit with the rest of the discussion and creates more confusion than clarity. Our aim here was to lay out some more of the potential benefits that a variety of differentiated social relationships may provide to humans (e.g., support of human life history trajectories, collaborative foraging niche). We wanted to mention those not to imply that our results support or negate this, but instead, we intended to introduce new avenues for potential future research in non-human animals. For example, if social relationships may promote group-level cooperation then group hunting in chimpanzees might be more successful if social bond partners or maternal kin jointly hunt together, but this remains to be explored. We have now clarified that we are not suggesting that the same benefits known to occur in chimpanzees or other non-human animals, but that this can be an avenue for future research (please see lines 282-289).

References:

- Apicella, C. L., Marlowe, F. W., Fowler, J. H., & Christakis, N. A. (2012). Social networks and cooperation in hunter-gatherers. *Nature*, *481*(7382), 497–501. <https://doi.org/10.1038/nature10736>
- Arseneau-Robar, T. J. M., Taucher, A. L., Schnider, A. B., van Schaik, C. P., & Willems, E. P. (2017). Intra- and interindividual differences in the costs and benefits of intergroup aggression in female vervet monkeys. *Animal Behaviour*, *123*, 129–137. <https://doi.org/10.1016/j.anbehav.2016.10.034>
- Cooper, M. A., Aureli, F., & Singh, M. (2004). Between-group encounters among bonnet macaques (*Macaca radiata*). *Behavioral Ecology and Sociobiology*, *56*(3), 217–227. <https://doi.org/10.1007/s00265-004-0779-4>
- Cords, M. (2007). Variable Participation in the Defense of Communal Feeding Territories by Blue Monkeys in the Kakamega Forest, Kenya. *Behaviour*, *144*(12), 1537–1550. JSTOR.
- Crockford, C., Wittig, R. M., Langergraber, K., Ziegler, T. E., Zuberbühler, K., & Deschner, T. (2013). Urinary oxytocin and social bonding in related and unrelated wild chimpanzees. *Proceedings of the Royal Society B: Biological Sciences*, *280*(1755), 20122765. <https://doi.org/10.1098/rspb.2012.2765>
- Dyble, M., Thompson, J., Smith, D., Salali, G. D., Chaudhary, N., Page, A. E., Vinicuis, L., Mace, R., & Migliano, A. B. (2016). Networks of Food Sharing Reveal the Functional Significance of Multilevel Sociality in Two Hunter-Gatherer Groups. *Current Biology*, *26*(15), 2017–2021. <https://doi.org/10.1016/j.cub.2016.05.064>
- Kitchen, D. M., Cheney, D. L., & Seyfarth, R. M. (2004). Factors Mediating Inter-Group Encounters in Savannah Baboons (*Papio cynocephalus ursinus*). *Behaviour*, *141*(2), 197–218. JSTOR.
- Koch, F., Signer, J., Kappeler, P. M., & Fichtel, C. (2016). Intergroup encounters in Verreaux's sifakas (*Propithecus verreauxi*): Who fights and why? *Behavioral Ecology and Sociobiology; Heidelberg*, *70*(5), 797–808. <http://dx.doi.org.ezp-prod1.hul.harvard.edu/10.1007/s00265-016-2105-3>
- Langergraber, K. E., Watts, D. P., Vigilant, L., & Mitani, J. C. (2017). Group augmentation, collective action, and territorial boundary patrols by male chimpanzees. *Proceedings of the National Academy of Sciences*, *114*(28), 7337–7342. <https://doi.org/10.1073/pnas.1701582114>
- Lucchesi, S., Cheng, L., Janmaat, K., Mundry, R., Pisor, A., & Surbeck, M. (2020). Beyond the group: How food, mates, and group size influence intergroup encounters in wild bonobos. *Behavioral Ecology*, *31*(2), 519–532. <https://doi.org/10.1093/beheco/arz214>
- Majolo, B., deBortoli Vizioli, A., Martínez-Íñigo, L., & Lehmann, J. (2020). Effect of Group Size and Individual Characteristics on Intergroup Encounters in Primates. *International Journal of Primatology*. <https://doi.org/10.1007/s10764-019-00119-5>
- Majolo, B., Ventura, R., & Koyama, N. F. (2005). Sex, Rank and Age Differences in the Japanese Macaque (*Macaca fuscata yakui*) Participation in Inter-Group Encounters. *Ethology*, *111*(5), 455–468. <https://doi.org/10.1111/j.1439-0310.2005.01087.x>
- Mirville, M. O., Ridley, A. R., Samedi, J. P. M., Vecellio, V., Ndagijimana, F., Stoinski, T. S., & Grueter, C. C. (2018). Factors influencing individual participation during intergroup

- interactions in mountain gorillas. *Animal Behaviour*, 144, 75–86.
<https://doi.org/10.1016/j.anbehav.2018.08.003>
- Nakagawa, S., & Schielzeth, H. (2013). A general and simple method for obtaining R² from generalized linear mixed-effects models. *Methods In Ecology And Evolution*, 4(2).
<https://pub.uni-bielefeld.de/publication/2565368>
- Nunn, C. L., & Deaner, R. O. (2004). Patterns of participation and free riding in territorial conflicts among ringtailed lemurs (*Lemur catta*). *Behavioral Ecology and Sociobiology*, 57(1), 50–61. <https://doi.org/10.1007/s00265-004-0830-5>
- Sakamaki, T., Ryu, H., Toda, K., Tokuyama, N., & Furuichi, T. (2018). Increased Frequency of Intergroup Encounters in Wild Bonobos (*Pan paniscus*) Around the Yearly Peak in Fruit Abundance at Wamba. *International Journal of Primatology*, 39(4), 685–704.
<https://doi.org/10.1007/s10764-018-0058-2>
- Samuni, L., Preis, A., Mielke, A., Deschner, T., Wittig, R. M., & Crockford, C. (2018). Social bonds facilitate cooperative resource sharing in wild chimpanzees. *Proc. R. Soc. B*, 285(1888), 20181643. <https://doi.org/10.1098/rspb.2018.1643>
- Silk, J. B., Beehner, J. C., Bergman, T. J., Crockford, C., Engh, A. L., Moscovice, L. R., Wittig, R. M., Seyfarth, R. M., & Cheney, D. L. (2009). The benefits of social capital: Close social bonds among female baboons enhance offspring survival. *Proceedings of the Royal Society B: Biological Sciences*, 276(1670), 3099–3104.
<https://doi.org/10.1098/rspb.2009.0681>
- Watts, D. P., & Mitani, J. C. (2001). Boundary patrols and intergroup encounters in wild chimpanzees. *Behaviour*, 138(3), 299–327. <https://doi.org/10.1163/15685390152032488>
- Wilson, M. L., Hauser, M. D., & Wrangham, R. W. (2001). Does participation in intergroup conflict depend on numerical assessment, range location, or rank for wild chimpanzees? *Animal Behaviour*, 61(6), 1203–1216. <https://doi.org/10.1006/anbe.2000.1706>
- Wittig, R. M., Crockford, C., Weltring, A., Langergraber, K. E., Deschner, T., & Zuberbühler, K. (2016). Social support reduces stress hormone levels in wild chimpanzees across stressful events and everyday affiliations. *Nature Communications*, 7, 13361.
<https://doi.org/10.1038/ncomms13361>

Reviewers' Comments:

Reviewer #1:

Remarks to the Author:

You have addressed the issues raised by reviewers to my satisfaction.

Some minor points:

85: "we used 58 observation years of demographic, genetic and behavioural data"

Would be useful here to state that these 58 years of data are from three communities (to clarify that data collection at Taï didn't begin in the 1960s...).

113: "a son approximating reproductive age"

170: "a son approximating reproductive age (>8 yrs)"

442: "sons approximating reproductive age"

"approaching" would be better than "approximating" here (given that age of first reproduction for males is usually much later than 8 years old — approaching gives a better sense of what is meant here).

292-293: "the evolutionary processes leading to human unique large-scale cooperation"

"human unique" is awkward. "uniquely human" seems to be what is meant here. Something like "distinctively human" would be better, given that social insects have cooperation on scales that rival humans, but achieve that cooperation through different mechanisms.

As an aside, I would take issue with the claim in your response to reviewers that "The collective action problem and 'strength in numbers' are therefore independent mechanisms." On the contrary, strength in numbers can provide a solution to the collective action problem: the presence of many allies reduces the costs of individual action, by making participation in an intergroup encounter less risky. Strength in numbers therefore increases the likelihood that any given individual will participate in producing the public good (in this case, giving a vocal response and/or approaching the rival community members).

Reviewer #3:

Remarks to the Author:

I was not one of the original reviewers for this paper, so was reading it for the first time along with the reviewers' comments and authors' responses. Overall, I think the paper is clearly written, tackles a timely topic with an excellent dataset and careful analyses, and will make a valuable contribution to the literature. In general, I think the authors have done a very good job in responding to the reviewers' comments. However, I do have a couple of points relating to those comments/responses, as well as a few issues of my own. These mostly relate to clarity and so I think all can be resolved with some rewriting and additions; none of them are fundamental problems.

General comments

Personally, I do not think the authors do themselves any favours by coming at topic of study from a human-behaviour standpoint – by which I mean, setting things up as if an important goal is to explain human behaviour. What that results in is a narrower-than-ideal focus on the non-human animal literature. This was something that was picked up on by the original reviewers and has clearly been improved in this version of the MS to some extent. But, there remain issues (i.e. that in places the MS focuses too much on just chimps or just primates, because the goal – even if only implicit – is to link to human behaviour). Enduring and strong social bonds are found in a wide range of taxa, and intergroup conflict is prevalent from ants to primates. I am not advocating a change in structure for this paper (though it is something to keep in mind for future papers), but I do think it would be useful

to widen the background in two main ways.

First, and specifically related to the point above, all elements of the first paragraph of the Introduction (which should be as broad as possible) ought to include reference to a wide range of species (not just primates). This is true in some sentences, but not others – most notably lines 35-37. A narrowing to chimps is probably fine from the following para, but I think the first paragraph should be as broad as possible.

Second, I think there would be value in adding a few sentences on how strong social bonds have been shown to influence cooperative behaviour. I fully agree with the authors that there has not been such consideration of their impact on the collective behaviour of intergroup conflict involvement. However, there are studies showing how strong bonds can affect decision / behaviour within groups (e.g. Schel et al. 2013 PLoS ONE 8: e76684; Fuong et al. 2015 Behav. Evol. 26: 587-592; Kern & Radford 2016 Biol. Lett. 12: 20160648). Including reference to those would in no way diminish the value of the current work, but would provide a fuller background of relevance.

Another general point relates to something that Reviewer 2 raised several times – the mention of “higher level structures of a social network”. Again, the authors have done a good job of resolving this specific issue. But I think part of the problem (which remains) is in using the term ‘social networks’. Lots of readers will immediately assume that the paper contains formal social-network analyses, which it does not. So, perhaps some confusion is created by including this phrase; if it is not crucial, then I would consider replacing with something else that is more clearly and instantly reflective of what has been measured and considered here.

I confess I am also slightly confused by the choice made with respect to the definition of “potential participants” and “participating individuals”. The latter is defined as those individual present in the current party when the encounter started who approached towards, vocalized and/or attacked the outgroup. That makes perfect sense to me. However, the “potential participants” category is any living adult in the full “community”. I.e. it could be individuals that are nowhere near the current encounter. Consequently, they cannot possibly be an actual participant, so I think some justification for using this rather than those who were at the encounter but did not approach, vocalize or attack is needed.

Finally, I think some care is needed about the use of the term “decision” in the paper. As a naïve reader, I might assume that an individual deciding whether to participate in an intergroup encounter depending on whom else is participating is doing so sequentially. I.e. that certain individuals have approached/vocalized/attacked the outgroup rivals and then what is considered is whom else joins in. However, I suspect that is not what is being analysed – rather, that any individual participating at any point in the intergroup encounter is considered a participant, and the analyses look at how that relates to other participants from any point in the encounter. That is not an issue itself, but the terminology of “decision-making” in a sequential sense might be a little misleading.

Specific comments

Lines 20–22: This tells the reader which factors were important but gives no hint as to the direction of effect. It would be more useful/powerful to provide the directional results in the Abstract.

Lines 22–24: I don't think the results of this study support the first part of this statement. You don't look at the benefits of maintaining a range of differential social relationships – you look at responses depending on their presence and find a benefit of having strong relationships. So some rewording advisable here.

Lines 73–77: This statement about revealing mechanisms is rather vague – it is unclear how examining your hypotheses can achieve this. I would either delete or make more explicitly clear how this can be so.

Table 2 (and associated results): Why is type of interaction not seemingly included in the male model, whereas it is for the female one in Table 1?

Figure 1: Why are the 'model' lines in panels B and C entirely below all the data points?

Line 273: You should specify that the male-biased predisposition for intergroup conflict participation is referring to chimps, as that is not the case in all other species.

Lines 313–318: Why do patrols not carry a direct risk; is there no chance of encountering rivals when on patrol? If not, why has previous work found that males are more likely to patrol with kin or those with whom they groom a lot (lines 68–70).

Lines 374–375: But if A strongly grooms B, would you not expect A to support B in intergroup interactions, even if the reverse were not true?

Line 384: A year is a very long time in the sense that social relationships could presumably change a lot in that period. So, is there any concern that, for example, two individuals could have a strong relationship in the first 6 months but it is now waning, yet they would have a similar score to two individuals who had a relatively weak initial relationship that has been getting much stronger recently?

We are grateful both reviewers for their valuable and positive feedback. Their comments have helped shape and improve this study.

Reviewer #1 (Remarks to the Author):

You have addressed the issues raised by reviewers to my satisfaction.

We thank the reviewer for investing their time again in reviewing our manuscript.

Some minor points:

85: "we used 58 observation years of demographic, genetic and behavioural data"
Would be useful here to state that these 58 years of data are from three communities (to clarify that data collection at Taï didn't begin in the 1960s...).

We have followed the suggestion of the reviewer. The sentence now reads "we used 58 **cumulative** observation years of demographic, genetic and behavioural data on 36 males and 75 females **from three groups** within the Taï Forest" (lines 92-93).

113: "a son approximating reproductive age"

170: "a son approximating reproductive age (>8 yrs)"

442: "sons approximating reproductive age"

"approaching" would be better than "approximating" here (given that age of first reproduction for males is usually much later than 8 years old — approaching gives a better sense of what is meant here).

We agree with the reviewer and implemented their suggestion

292-293: "the evolutionary processes leading to human unique large-scale cooperation"

"human unique" is awkward. "uniquely human" seems to be what is meant here. Something like "distinctively human" would be better, given that social insects have cooperation on scales that rival humans, but achieve that cooperation through different mechanisms.

Per the reviewer's suggestion we replaced the term 'uniquely human' by 'distinctively human'

As an aside, I would take issue with the claim in your response to reviewers that "The collective action problem and 'strength in numbers' are therefore independent mechanisms." On the contrary, strength in numbers can provide a solution to the collective action problem: the presence of many allies reduces the costs of individual action, by making participation in an intergroup encounter less risky. Strength in numbers therefore increases the likelihood that any given individual will participate in producing the public good (in this case, giving a vocal response and/or approaching the rival community members).

We fully agree with the reviewer that strength in numbers can provide a solution to collective action by reducing the costs of the risky act and promoting participation, and that referring to the two as 'independent mechanisms' may be misleading.

By stating that those are 'independent mechanisms' in the previous responses to the reviewers we did not mean to say that there is no connection between the two or that one cannot regulate the other. Instead we meant that each could occur with or without the other. For example, strength in numbers is an effective means of promoting participation, but when there is a large imbalance in the sizes of neighboring groups the larger group can likely achieve numerical superiority with many free riders thus still suffering from collective action problem.

Reviewer #3 (Remarks to the Author):

I was not one of the original reviewers for this paper, so was reading it for the first time along with the reviewers' comments and authors' responses. Overall, I think the paper is clearly written, tackles a timely topic with an excellent dataset and careful analyses, and will make a valuable contribution to the literature. In general, I think the authors have done a very good job in responding to the

reviewers' comments. However, I do have a couple of points relating to those comments/responses, as well as a few issues of my own. These mostly relate to clarity and so I think all can be resolved with some rewriting and additions; none of them are fundamental problems.

We are grateful to the reviewer for agreeing to review this manuscript and for their constructive feedback which helped improve the clarity and strength of our study.

General comments

Personally, I do not think the authors do themselves any favours by coming at topic of study from a human-behaviour standpoint – by which I mean, setting things up as if an important goal is to explain human behaviour. What that results in is a narrower-than-ideal focus on the non-human animal literature. This was something that was picked up on by the original reviewers and has clearly been improved in this version of the MS to some extent. But, there remain issues (i.e. that in places the MS focuses too much on just chimps or just primates, because the goal – even if only implicit – is to link to human behaviour). Enduring and strong social bonds are found in a wide range of taxa, and intergroup conflict is prevalent from ants to primates. I am not advocating a change in structure for this paper (though it is something to keep in mind for future papers), but I do think it would be useful to widen the background in two main ways.

We appreciate the reviewer's comments and have broadened the background and parts of the discussion as to present a wider view of social relationships and intergroup competition across the animal kingdom. We provide specific replies to the two main comments related to this topic below.

First, and specifically related to the point above, all elements of the first paragraph of the Introduction (which should be as broad as possible) ought to include reference to a wide range of species (not just primates). This is true in some sentences, but not others – most notably lines 35-37. A narrowing to chimps is probably fine from the following para, but I think the first paragraph should be as broad as possible.

To avoid providing a primate-centric view in the beginning of the introduction, we have added more details and citations to make the point that a wide range of social relationships, including social bonds, are evident across animal taxa.

Second, I think there would be value in adding a few sentences on how strong social bonds have been shown to influence cooperative behaviour. I fully agree with the authors that there has not been such consideration of their impact on the collective behaviour of intergroup conflict involvement. However, there are studies showing how strong bonds can affect decision / behaviour within groups (e.g. Schel et al. 2013 PLoS ONE 8: e76684; Fuong et al. 2015 Behav. Evol. 26: 587-592; Kern & Radford 2016 Biol. Lett. 12: 20160648). Including reference to those would in no way diminish the value of the current work, but would provide a fuller background of relevance.

We are grateful to the reviewer for pointing this out. Following their suggestion, we have incorporated the role of differentiated social relationships in promoting within group dyadic cooperation that is associated with fitness benefits, like in the context of predator/threat defense (as the reviewer suggested) and alloparental care (lines 42-44).

Another general point relates to something that Reviewer 2 raised several times – the mention of “higher level structures of a social network”. Again, the authors have done a good job of resolving this specific issue. But I think part of the problem (which remains) is in using the term ‘social networks’. Lots of readers will immediately assume that the paper contains formal social-network analyses, which it does not. So, perhaps some confusion is created by including this phrase; if it is not crucial, then I would consider replacing with something else that is more clearly and instantly reflective of what has been measured and considered here.

As we agree with the reviewer that the term “social networks” may lead to ambiguity, we now instead use more explicit terms like “range of differentiated relationships”.

I confess I am also slightly confused by the choice made with respect to the definition of “potential participants” and “participating individuals”. The latter is defined as those individual present in the

current party when the encounter started who approached towards, vocalized and/or attacked the outgroup. That makes perfect sense to me. However, the “potential participants” category is any living adult in the full “community”. I.e. it could be individuals that are nowhere near the current encounter. Consequently, they cannot possibly be an actual participant, so I think some justification for using this rather than those who were at the encounter but did not approach, vocalize or attack is needed.

We appreciate the comment by the reviewer and the opportunity to explain our approach in more detail. We defined “potential participants” as any living adult in the community to investigate group-level variation in participation tendencies. This definition followed a previous study in chimpanzees (Langregraber et al., 2017) that looked at participation of males in border patrols. By following the same definition for “potential participants” as previously used in chimpanzees we facilitate comparative assessments of participation drivers. We hope that the reviewer appreciates this perspective.

The reasoning behind this approach, is that it allows us to investigate between-individual variation in costs and benefits that can influence participation decisions in intergroup encounters without making assumptions on who was knowledgeable of the encounter or when the knowledge of an imminent encounter is acquired. This is a non-trivial point in a dense forest environment with a fission-fusion species for which buttress drumming can be heard over 1.5 km away. Also, intergroup contest can occur repeatedly over the course of a couple of weeks at the same fruiting tree. Cautious, silent approach to these trees in days following an encounter suggests that chimpanzees remember previous encounters. Thus, we cannot rule out that non-present chimpanzees are avoiding a current intergroup threat.

Thus, although the reviewer is correct that participation in intergroup encounter can be influenced by short-term decisions occurring just before or as the encounter starts, participation is also heavily influenced by longer-term decisions of individuals. For example, intergroup encounters in chimpanzees occur in specific peripheral areas of the territory. As such, group members can strategize where they range within the territory so to reduce the likelihood to encounter neighbors. Individuals can decide for example to forage mainly in core areas where the risk of encounters is minimized. In such case, encounter avoidance can happen long before encounters occur. Conversely, other individuals may be more likely to travel towards and forage in peripheral areas where it is potentially riskier. Since group movement towards peripheral areas can occur hours or days before the encounter, it is challenging defining the specific point where individuals make the decision to suffer the potential costs of encountering rivals.

The assumption is that individuals differ in how costly intergroup encounters may be to them and in what they can gain from successful competition with neighboring groups. For example, female chimpanzees in some population often remain within core areas and rarely participate in intergroup encounters. Although they may not be knowledgeable of the timing of encounters, they nonetheless make the active decision to not be around peripheral areas. Altogether, by limiting the data to the individuals present around the encounter areas we are likely to eliminate valuable information that may inform us on between individual variation in participation tendencies. As we are interested in detecting the patterns that predict participation tendencies, it is informative to include all adult individuals as potential participants. We have clarified this in the manuscript in lines 403-406.

Finally, I think some care is needed about the use of the term “decision” in the paper. As a naïve reader, I might assume that an individual deciding whether to participate in an intergroup encounter depending on whom else is participating is doing so sequentially. I.e. that certain individuals have approached/vocalized/attacked the outgroup rivals and then what is considered is whom else joins in. However, I suspect that is not what is being analysed – rather, that any individual participating at any point in the intergroup encounter is considered a participant, and the analyses look at how that relates to other participants from any point in the encounter. That is not an issue itself, but the terminology of “decision-making” in a sequential sense might be a little misleading.

We understand the reviewers’ point and acknowledge that the term ‘decision-making’ may be misleading in some of the cases. To prevent ambiguity, we avoid the use of this term except for a single case where it is clear that we are not inferring a sequential process: “we could use the identity

of individuals present during the encounter as a proxy of a pre-encounter decision-making process" (line 91).

Specific comments

Lines 20–22: This tells the reader which factors were important but gives no hint as to the direction of effect. It would be more useful/powerful to provide the directional results in the Abstract.

We have revised the sentence to be more explicit about the direction of the effect: "We found that participation increased with a greater number of other participants as well as when participants were maternal kin or social bond partners" (lines 24-25).

Lines 22–24: I don't think the results of this study support the first part of this statement. You don't look at the benefits of maintaining a range of differential social relationships – you look at responses depending on their presence and find a benefit of having strong relationships. So some rewording advisable here.

We see the reviewer's point and have revised the sentence to be clear that we do not look at the benefits stemming from the maintenance of a range of social relationships, but instead we examine how having those relationships may facilitate cooperation during intergroup encounters (line 27).

Lines 73–77: This statement about revealing mechanisms is rather vague – it is unclear how examining your hypotheses can achieve this. I would either delete or make more explicitly clear how this can be so.

To avoid unclarity, we have expanded the sentence to include the potential mechanisms we investigate that may aid chimpanzees to reduce the costs and increase the benefits of intergroup interactions. Specifically, we note in lines 80-84 that: "examining the hypotheses that the number of participants and social preference influence the decision to take part in hostile encounters with neighbours may potentially reveal whether numerical strength and/or partner choice are mechanisms through which chimpanzees optimise the cost-to-benefit ratio associated with such conflicts to promote cooperation"

Table 2 (and associated results): Why is type of interaction not seemingly included in the male model, whereas it is for the female one in Table 1?

The type of interaction was accidentally omitted from Table 2 and the result section but was included in the 'male model' that we fitted, we have rectified this mistake.

Figure 1: Why are the 'model' s in panels B and C entirely below all the data points?

We thank the reviewer for spotting this. We investigated the issue to find that the model line appeared below the datapoint because we did not account for the offset term properly when plotting the data. We have corrected this and provide updated plots, but in one case the model line still appears on the lower side of the points (likely due to the effect of the offset term).

Line 273: You should specify that the male-biased predisposition for intergroup conflict participation is referring to chimps, as that is not the case in all other species.

We have clarified that male-biased predisposition for intergroup conflict participation is evident in some primate species, including chimpanzees and humans (lines 258-259)

Lines 313–318: Why do patrols not carry a direct risk; is there no chance of encountering rivals when on patrol? If not, why has previous work found that males are more likely to patrol with kin or those with whom they groom a lot (lines 68–70).

The reviewer is correct that border patrols carry a risk in case rival chimpanzees are encountered. For that reason, we focused on all intergroup events that included interactions with rivals most of which consisted of a border patrol behavior. Nonetheless, as we agree that our wordings may cause confusion, we have edited the wordings in the sentence to provide more clarity (please see line 309).

Lines 374–375: But if A strongly grooms B, would you not expect A to support B in intergroup interactions, even if the reverse were not true?

This is an interesting point raised by the reviewer. Unbalanced grooming in chimpanzees can be a characteristic of different types of relationships and often fluctuates through time, thus it is associated with high uncertainty. For example, low ranking or young males are likely to invest more in the grooming of high ranking or older males than vice versa, and male chimpanzees may groom receptive females but will receive little back. In both examples, partner choice is expected to vary according to the social or reproductive status of partners and as a function of partner availability, and thus be unpredictable. As we were interested in how predictability of support may influence participation in intergroup interactions, we focused on social relationships that are mutual and stable as those are the types of relationships associated with high consistency of interactions through time.

Further, since our data points in the model are individual chimpanzees and not dyads, our predictors are either at the level of the individual (e.g., age, dominance rank) or the group (e.g., adult kin/bond partner present, number of male/female participants). Therefore, at its current structure, it is impossible to investigate directional dyadic parameters (e.g., grooming strength from A to B) in the model.

Line 384: A year is a very long time in the sense that social relationships could presumably change a lot in that period. So, is there any concern that, for example, two individuals could have a strong relationship in the first 6 months but it is now waning, yet they would have a similar score to two individuals who had a relatively weak initial relationship that has been getting much stronger recently?

The reviewer is correct that social relationships, or here specifically association relationships, can fluctuate throughout the year on a dyadic level. However, association tendencies as we measured them (i.e., on the group level as a measure of gregariousness) are highly repeatable on a daily and yearly basis in Tai chimpanzees (Tkaczynski et al., 2020). Here, we were not interested in the strength of dyadic association relationships but instead we wanted to control in the statistical analysis for the likelihood that more gregarious individuals are more likely to be present in the party and thus may be more likely to participate. For that reason, we calculated association patterns as the mean dyadic association index an individual had with all other participants.

When assessing the association tendencies of individuals within a group, we needed to find the balance between having enough data to construct reliable estimates and using a timeframe that encompasses environmental changes that have a strong impact on gregariousness. Therefore, the 1-yr timeframe provided sufficient data to construct the association index but also accounted for annual seasonal patterns in food availability, temperature, and rainfall that influence association patterns. Another aspect known to affect association tendencies, the demography of the group (e.g., group size, sex ratio) is also expected to be fairly stable in a 1-yr timeframe. Thus, although dyadic association patterns may fluctuate within a year leading to some measurement error, the average association value over a year likely provides a better overall estimate of the social landscape of individuals within the group. Given that we found a strong positive relationship between the average association values and participation likelihood suggests that our measure of association tendencies is meaningful.

To clarify this, we have added information on repeatability in association patterns in males and females in Tai indicating stable individual differences in gregariousness (lines 374-376).

Reviewers' Comments:

Reviewer #3:

Remarks to the Author:

I was Reviewer #3 for the previous submission of this paper. I think that the authors have done a generally excellent job of responding to my own comments and to those of the other reviewer. Accordingly, I have no major issues remaining and certainly nothing relating to the data collection, analyses or big conclusions, and believe this will be a strong addition to the literature.

I just have just one general point for the authors to ponder in this final revision (at most this just requires a little tweaking of language). Perhaps the most novel and interesting finding of the paper is that chimps were more likely to participate in intergroup encounters when other participants were those with whom they are strongly bonded. This is set up well in lines 66-85, where the wording used is (correctly in my mind) about "strong" relationships/bonds, and there are some clear findings in this regard. What confuses me in other parts of the paper, including in the Abstract (lines 26-27) and elements of the Discussion, is the claim that the "variety" of social relationships facilitates group-level cooperation. The implication seems to be that having relationships of different strengths with different individuals aids collective cooperation in intergroup conflict. Maybe I am missing something, but this doesn't make immediate sense to me – why does having some weak bonds (as well as strong bonds) facilitate this; surely it is having strong bonds that matters (at least in terms of what you have shown with your data). If the relevant parts of the paper were worded along the lines of, "enduring, strong bonds facilitate group-level cooperation", that would seem to tie directly to the results and be a compelling, clear conclusion. Forgive me if I have misunderstood, but I think this just needs a bit of thought for complete clarity.

As a minor point, it seems slightly odd (unless this is a journal requirement) to have the final paragraph of the Introduction (lines 118-125) as a summary of the main findings before we get to the Results. That paragraph would seem more appropriately placed at the beginning of the Discussion.

I was Reviewer #3 for the previous submission of this paper. I think that the authors have done a generally excellent job of responding to my own comments and to those of the other reviewer. Accordingly, I have no major issues remaining and certainly nothing relating to the data collection, analyses or big conclusions, and believe this will be a strong addition to the literature.

We are grateful to the reviewer for the valuable feedback they provided in both rounds of revision.

I just have just one general point for the authors to ponder in this final revision (at most this just requires a little tweaking of language). Perhaps the most novel and interesting finding of the paper is that chimps were more likely to participate in intergroup encounters when other participants were those with whom they are strongly bonded. This is set up well in lines 66-85, where the wording used is (correctly in my mind) about strong relationships/bonds, and there are some clear findings in this regard. What confuses me in other parts of the paper, including in the Abstract (lines 26-27) and elements of the Discussion, is the claim that the variety of social relationships facilitates group-level cooperation. The implication seems to be that having relationships of different strengths with different individuals aids collective cooperation in intergroup conflict. Maybe I am missing something, but this doesn't make immediate sense to me why does having some weak bonds (as well as strong bonds) facilitate this; surely it is having strong bonds that matters (at least in terms of what you have shown with your data). If the relevant parts of the paper were worded along the lines of enduring, strong bonds facilitate group-level cooperation that would seem to tie directly to the results and be a compelling, clear conclusion. Forgive me if I have misunderstood, but I think this just needs a bit of thought for complete clarity.

We thank the reviewer for pointing this out and agree that the phrase "variety of social relationships" is confusing. We have followed their suggestion and clarified in both the abstract and discussion that it is strong social bonds that facilitated group-level cooperation in chimpanzees.

As a minor point, it seems slightly odd (unless this is a journal requirement) to have the final paragraph of the Introduction (lines 118-125) as a summary of the main findings before we get to the Results. That paragraph would seem more appropriately placed at the beginning of the Discussion.

We have followed the journal guidelines which require that the last paragraph of the introduction contain a summary of the findings and conclusions of the current work written in present tense.